# FreeControl: Efficient, Training-Free Structural Control via One-Step Attention Extraction

**Jiang Lin**[1,†], **Xinyu Chen**[1], **Song Wu**[2], **Zhiqiu Zhang**[1], **Jizhi Zhang**[1], **Ye Wang**[4],
**Qiang Tang**[3], **Qian Wang**[2], **Jian Yang**[1], **Zili Yi**[1*]

[1]Nanjing University, Suzhou, China
[2] JIUTIAN Research, Beijing, China
[3]University of British Columbia, Vancouver, Canada
[4]Jilin University, Changchun, China

lin@smail.nju.edu.cn, yi@nju.edu.cn

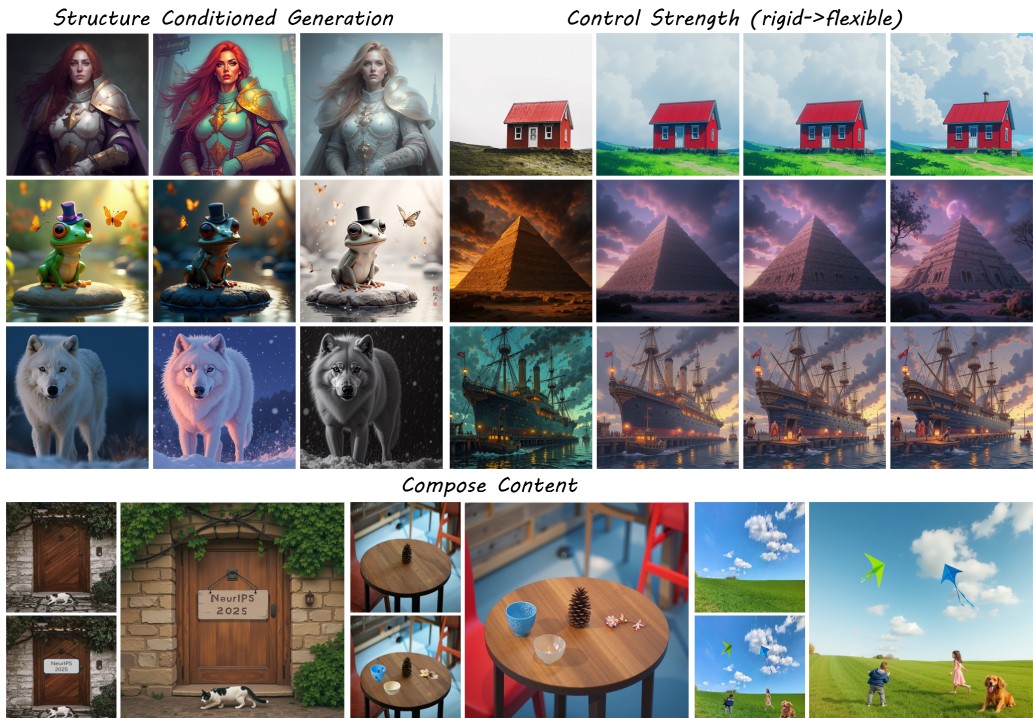

Figure 1: FreeControl enables efficient, structure-aware generation from raw image references. Top-left: structure-conditioned generation using reference image on the left. Top-right: Tunable Control strength via adjustable attention injection. Bottom: compositional generation from user-assembled reference images enables intuitive spatial and semantic layout control.

## Abstract

Controlling the spatial and semantic structure of diffusion-generated images remains a challenge. Existing methods like ControlNet rely on handcrafted condition maps and retraining, limiting flexibility and generalization. Inversion-based approaches offer stronger alignment but incur high inference cost due to dual-path denoising. We present **FreeControl**, a training-free framework for semantic struc-

*Corresponding authors: yi@nju.edu.cn; †: project lead

39th Conference on Neural Information Processing Systems (NeurIPS 2025).

tural control in diffusion models. Unlike prior methods that extract attention across multiple timesteps, FreeControl performs *one-step attention extraction* from a single, optimally chosen key timestep and reuses it throughout denoising. This enables efficient structural guidance without inversion or retraining. To further improve quality and stability, we introduce *Latent-Condition Decoupling (LCD)*: a principled separation of the key timestep and the noised latent used in attention extraction. LCD provides finer control over attention quality and eliminates structural artifacts. FreeControl also supports compositional control via reference images assembled from multiple sources, enabling intuitive scene layout design and stronger prompt alignment. FreeControl introduces a new paradigm for test-time control—enabling structurally and semantically aligned, visually coherent generation directly from raw images, with the flexibility for intuitive compositional design and compatibility with modern diffusion models at ~5% additional cost.

# 1    Introduction

While diffusion models [8, 33, 36, 32, 5, 28, 2] have revolutionized generative image synthesis, they remain difficult to control. There often lacks intuitive ways to specify what appears where, or how objects should relate spatially and semantically—making them less suitable for tasks like scene layout, object rearrangement, or design prototyping.

A prevalent approach to this challenge involves conditioning generation on external control maps, as exemplified by ControlNet [47] and T2I-Adapter [24]. These methods inject spatial guidance via edge maps, depth cues, or segmentation masks. While effective, they depend on handcrafted pre-processing and require separate training for each control type and base model. In particular, ControlNet demands large-scale paired datasets and substantial training resources per condition, making it expensive to scale across modalities or architectures. Moreover, the control signals themselves are limited: Canny edges are often overly rigid and may conflict with prompt semantics, while segmentation-based guidance is restricted by limited category labels, preventing nuanced or open-ended structure control. In contrast, test-time augmented methods [38, 23, 19, 40] such as DDIM inversion [38] and TRAC [19] extract structure from reference images by reconstructing their latent trajectory and injecting features throughout denoising. These techniques incur high inference cost, requiring full or dual-path denoising and considerable memory.

We propose a training-free, test-time augmented framework for semantic structural control using raw reference images. Our method performs a single additional denoising step at a model-specific key timestep, chosen to extract maximally informative self-attention. This attention matrix captures both spatial structure and semantic intent, and is consistently injected into the main generation process to guide the arrangement and content of the generated image. The strength and scope of guidance are tunable, enabling both flexible layout guidance and strong structural adherence, depending on user intent. Unlike prior test-time augmentation methods [7, 4], our approach eliminates the need for inversion or reconstruction entirely—removing both the computational burden and architectural complexity of dual-path denoising. With only 5% additional cost over baseline inference, it delivers high-quality structural control without retraining, making it directly compatible with fine-tuned [35] or LoRA-augmented models [10].

By collapsing multi-step extraction into a single attention signal, our one-step approach creates a tractable point of analysis—enabling us to systematically study and refine the quality of structural guidance through Latent-Condition Decoupling (LCD). LCD separates the roles of the noised latent and the key timestep, revealing how each factor shapes the extracted structure. This lets us improve alignment quality, reduce artifacts, and offer tunable control over structural granularity—from coarse layout to fine semantic detail.

To support intuitive, layout-aware control beyond segmentation maps or prompt tuning, we introduce a composition-based conditioning strategy. As shown in fig. 1, users can directly assemble reference images by cropping and combining objects from different sources, enabling them to express both spatial layout and semantic intent in a natural visual form, and generate images with content that aligns with their expectations. This flexibility transforms structural and semantic conditioning into a designable interface for high-level scene control.

In experiments, FreeControl outperforms existing structural control methods in both spatial alignment and visual fidelity, while maintaining high efficiency. Qualitative results further demonstrate its advantage in semantic-level control, producing generations that more faithfully adhere to the intended prompts.

Our contributions are as follows:

- We present a training-free, test-time augmented method for semantic structural control from raw reference images, eliminating the need for handcrafted inputs, inversion, or retraining.

- We propose a one-step attention extraction framework that uses a single denoising step at a key timestep to guide generation, with attention maps injected across layers during inference.

- We introduce Latent-Condition Decoupling (LCD), a principled method that separates the key timestep from the noised latent in attention extraction, enabling stronger control and improved stability.

- We introduce a composition-based conditioning approach that allows users to define both spatial layout and semantic intent through assembled reference images, enabling intuitive control beyond segmentation maps or prompt tuning.

## 2   Related Work

**Diffusion Models.** Diffusion models [8, 38, 5, 28] have emerged as a leading framework for high-quality image synthesis, with success across tasks such as text-to-image generation [33, 36, 32], image-to-image translation [12, 22, 15], and image editing [7, 3, 45, 13, 37]. Foundational models like DDPM [8] and DDIM [38] introduced the core denoising process, while later developments such as LDM [33], DiT [27], and SD3 [6] have scaled diffusion to high-resolution, semantically rich generation. The field has also moved from U-Net-based backbones [34] to more expressive transformer-based architectures [41].

**Structural Guidance via Training-Based Conditioning.** Training-based methods such as Control-Net [47] and T2I-Adapter [24] guide spatial structure using condition maps like edges or segmentation. While they achieve strong low-level alignment, they require retraining for each control type and base model—introducing high computational cost and model proliferation. Their reliance on handcrafted inputs also leads to brittle performance when structure maps are noisy or conflict with text prompts. T2I-Adapter is lighter but similarly struggles with complex scenes and still demands per-condition training. High-level alternatives like GLIGEN [18] and IP-Adapter [46] support layout-aware and visual conditioning using bounding boxes or global image features. However, IP-Adapter still needs base-model-specific training and shows varied quality depending on the dataset. Though these methods improve compositional flexibility, they lack fine-grained structural control—e.g., for object contours or pose—limiting their utility in dense structure transfer tasks.

**Test-Time Control via Inversion and Attention Reuse.** Test-time approaches offer another path to structure control. DDIM inversion [38, 40, 21] and Null-Text Inversion [23] reconstruct noise latents from reference images, enabling attention reuse for editing. Though effective, they are computationally heavy and depend on prompt alignment. Prompt-to-Prompt [7] modifies cross-attention for semantic edits while preserving layout but cannot incorporate visual references.

Plug-and-Play [40] injects image features during inference but offers coarse structure control. It requires dual attention modulation and ResNet backbones, limiting efficiency and compatibility with transformer-based models like DiTs [27]. TRAC [19] improves efficiency by avoiding inversion, but still extracts attention at many timesteps, incurring high cost.

Crucially, both inversion-based and inversion-free methods operate under the same assumption: that structure must be extracted progressively across a denoising trajectory. Yet, across timesteps, the role of attention remains consistent—capturing spatial layout and semantic structure. This raises a fundamental question: if attention serves the same purpose at every step, is repeated extraction truly necessary?

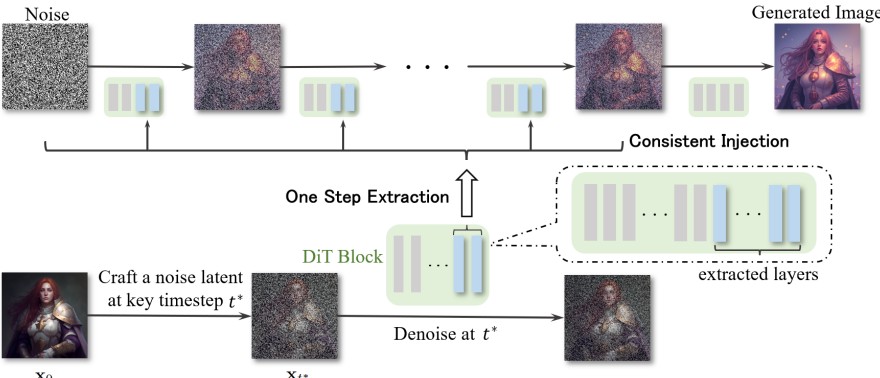

Figure 2: The illustration of one-step attention extraction framework. The query attention matrices in the later layers (blue layers) are extracted from a forward-simulated latent at a single key timestep and are injected consistently in the generation process to enable structural guidance.

## 3 Methods

### 3.1 Semantic Structural Control via One-Step Extraction

**One-Step Attention Extraction** Motivated by the insight that structural information remains conceptually consistent across timesteps, we introduce a *one-step attention extraction strategy* to replace multi-step guidance. As demonstrated in fig. 2, rather than accumulating structure through repeated attention capture, we extract attention matrices once from a single, designated timestep and reuse them throughout the denoising process. This approach preserves structural alignment while significantly reducing computational overhead.

A key design consideration in this framework is the selection of the optimal timestep $t^*$ for attention extraction. We conduct an empirical evaluation across a range of candidate steps and identify the one (661) that yields the strongest structural alignment in the final output. In contrast to inversion-based methods, which require traversing a full reverse denoising trajectory to reach $t^*$, we adopt a lightweight forward simulation strategy: we apply the forward noise process directly to the reference latent $\mathbf{x}_0$ to simulate the noised latent at timestep $t^*$, bypassing reverse diffusion entirely. Specifically, we compute the noised latent as:

$$\mathbf{x}_{t^*} = \sigma_{t^*} \cdot \boldsymbol{\epsilon} + (1 - \sigma_{t^*}) \cdot \mathbf{x}_0 \tag{1}$$

where $\boldsymbol{\epsilon} \sim \mathcal{N}(0, \mathbf{I})$ is standard Gaussian noise, and $\sigma_{t^*} \in [0, 1]$ is the timestep-dependent noise scale factor at optimal timestep $t^*$. A single denoising step is then applied to $\mathbf{x}_{t^*}$ to produce intermediate attention maps. From this, we extract the self-attention query matrices $\mathbf{Q}_{t^*}^{(l)}$ at each transformer layer $l$. During generation, these matrices are injected at every timestep $t$ by replacing the model's dynamically computed queries:

$$\mathbf{Q}_t^{(l)} \leftarrow \mathbf{Q}_{t^*}^{(l)} \tag{2}$$

The key ($\mathbf{K}$) and value ($\mathbf{V}$) matrices remain dynamically computed from the evolving latent $\mathbf{x}_t$, preserving responsiveness to the generative context while maintaining consistent structural queries. This procedure introduces no per-image tuning and requires only a single additional denoising step, making our method highly efficient, training-free, and broadly applicable across diffusion architectures.

**Layer-Aware Injection for Preserving Appearance Quality.** While one-step attention extraction provides effective structural control, indiscriminate injection across all layers can degrade visual quality. In particular, injecting structural $\mathbf{Q}$ matrices into early layers of the diffusion transformer interferes with low-level synthesis tasks—such as color, lighting, and texture modeling—often resulting in desaturation, flat textures, or unnatural shading. This occurs because early layers are primarily responsible for fine appearance features, and rigid structural guidance can disrupt their generative flexibility.

In contrast, deeper transformer layers capture higher-level semantic and spatial information, making them more suitable targets for structural injection. To balance structure and appearance, we adopt a layer-aware injection strategy that applies structural $\mathbf{Q}$ matrices only to mid-to-late layers (the blue layers in fig. 2). This preserves structure alignment while allowing early layers to focus on generating detailed and visually rich content.

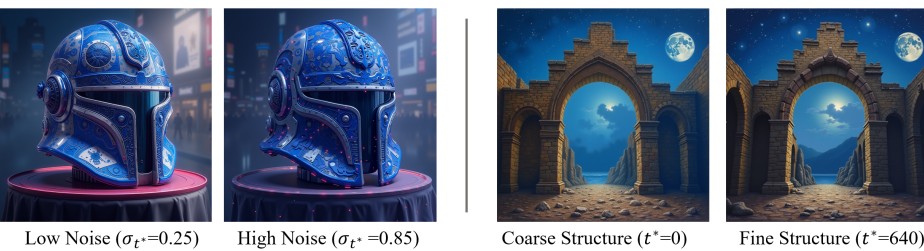

| Low Noise ($\sigma_{t^*}$=0.25) | High Noise ($\sigma_{t^*}$ =0.85) | Coarse Structure ($t^*$=0) | Fine Structure ($t^*$=640) |

Figure 3: Left: Noisy artifacts induced by the noise term. Right: Different granularity of structural control under different key timesteps.

## 3.2 Latent-Condition Decoupling (LCD) for Enhanced Attention Quality

With attention now extracted from a single forward-simulated timestep, we gain a stable and isolated point of intervention for improving control. We introduce *Latent-Condition Decoupling (LCD)* to exploit this opportunity. Rather than varying the key timestep $t^*$, LCD disentangles the two core influences on attention quality: (1) the noised latent $\mathbf{x}_{t^*}$ provided as input, and (2) the key timestep passed to the model. By isolating and manipulating these factors independently, we gain deeper insight into how structural guidance arises—and unlock both improved fidelity and fine-grained control.

To isolate the contribution of the noised latent, we fix the key timestep (at previously optimal 661) and vary the construction of $\mathbf{x}_{t^*}$. As shown in fig. 3, we find that latents generated with high noise levels (i.e., large $\sigma_{t^*}$) tend to introduce visible noise artifacts such as scattered dots in the final image. These high-noise latents also degrade the attention maps extracted for structural guidance, likely due to the model's inability to reason clearly over heavily corrupted input.

Based on this insight, we propose a simplified latent construction that removes the stochastic noise term entirely. Rather than performing forward diffusion with sampled noise, we directly construct a scaled $\tilde{\mathbf{x}}$, which serves as a substitute for $\mathbf{x}_{t^*}$.

$$\tilde{\mathbf{x}} = (1 - \sigma) \cdot \mathbf{x}_0,$$

where $\sigma$ here becomes a tunable scale factor, independent of the key timestep. The removal of the noise term improves the stability of the proposed method. This noise-free latent simulates the amplitude characteristics of an intermediate timestep while preserving the spatial coherence of the original image latent. Empirical evaluation shows that moderate values (e.g., $\sigma \in [0.25, 0.5]$) yield the best results.

We then fix the latent input $\tilde{\mathbf{x}}$ and vary the key timestep passed to the diffusion transformer. As shown in fig. 3, the choice of key timestep affects the granularity of structural control. Conditioning with a timestep near zero yields prominent but coarse structure—capturing large shapes and global layout while omitting fine detail. In contrast, using a timestep closer to the original key timestep (e.g., 661) achieves finer structure transfer, preserving contours, texture boundaries, and detailed object shapes.

This observation opens the door to user-driven structural tradeoffs: by adjusting the conditioning timestep, one can control the rigidity of structural guidance. Lower key timesteps provide more compositional flexibility—suitable for creative reinterpretations or stylistic variation—while higher key timesteps enforce stricter alignment with the reference structure. This tunable granularity makes LCD not only a tool for quality improvement, but also a mechanism for interactive control.

## 3.3 Unlocking the Full Potential of FreeControl with Compositional Generation

With the core attention control mechanism in place, FreeControl serves as a flexible framework for structural guidance, enabling users to intuitively define spatial and semantic layout without modifying

the model. Rather than relying on hand-crafted segmentation maps or prompt engineering, users can directly control both the content and position of visual elements through image composition.

**Compositional Reference Images for Semantic Layouts.** While structural control methods like ControlNet or T2I-Adapter are effective at enforcing edges or spatial layouts, they often struggle to preserve semantic intent—particularly in complex scenes. FreeControl addresses this by enabling compositional reference images, where users can define both *what* should appear and *where* it should appear using direct visual assembly.

For instance, a user can extract an object (e.g., a kite) from a source image using a segmentation tool like SAM [14] and paste it onto a new background (e.g., sky). The resulting image encodes both spatial structure and semantic intent, and serves as a direct condition for generation. This "design by composition" approach allows users to guide the layout in a natural, intuitive way—without requiring segmentation maps or prompt engineering. Examples are shown in fig. 1, which comprise cases such as transferring digital text to real writing and scene composition.

To improve robustness when assembling such references, Gaussian blur could be optionally applied to the compositional image before passing it to the model. This lightweight preprocessing step reduces sharp boundaries and high-frequency noise, helping the model focus on the intended structure while avoiding artifacts from copy-paste seams.

# 4 Experiments

## 4.1 Implementation details

We conduct all experiments using the FLUX.1-dev [16] model with the FlowMatchEulerDiscrete scheduler, a timestep range of 1000 to 400, and a guidance scale of 6.5. Quantitative results use 25 denoising steps; 50 steps are used elsewhere for improved visual quality. The key timestep $t^*$ is fixed at 661. Attention is extracted once and injected into the last 25 transformer layers of the model's single transformer block in the quantitative evaluations, and may be reduced elsewhere to demonstrate results of lower structural control. Compositional image generation is disabled unless specifically ablated. Inference is performed on a single NVIDIA RTX A6000 GPU with 48 GB of memory, and the inference time is measured over 100 runs.

## 4.2 Quatitative Comparison

**Dataset.** We evaluate on 5,000 images sampled from the COCO 2017 [20] validation set, resized to 512×512. Each image is paired with its corresponding caption, which is used as the input text prompt for controlled generation.

**Metrics.** We report **FID** for visual fidelity, **SSIM** and **PSNR** for low-level similarity, and **CLIP-Text Similarity** [31] for semantic alignment between images and prompts. For Canny-conditioned models, we quantify structural fidelity with the **F1 score** computed between the input Canny edge map and the Canny edge map extracted from each generated image. For depth-conditioned models, we report pixel level accuracy as the **mean squared error (MSE)** between the input depth map and the depth map predicted from the generated image.

**Comparison Methods.** We compare FreeControl against five strong baselines: ControlNet [47], UniControlNet [49], UniControl [30], ControlNet++ [17], and Flux-ControlNet [43, 44]. The first four baselines are implemented on Stable Diffusion v1.5, while Flux ControlNet is built on FLUX.1-dev[16]. All SD 1.5-based models are run with 20 denoising steps, and Flux-based methods—including FreeControl—use 25 steps, following the respective official configurations.

Note that ControlNet-style methods require pre-processed condition maps (e.g., Canny edge or depth), while FreeControl directly uses the raw image as structural input, with no preprocessing(we also did not count the preprocessing time for the comparison methods in table 2). The Canny edge is computed with the high threshold set to 200, and the low threshold set to 100.

**Results.** Table 1 reports quantitative comparisons across several metrics. FreeControl outperforms all baselines in terms of structural similarity (SSIM and PSNR), while maintaining competitive CLIP-Text alignment with prompt semantics. Compared to ControlNet and UniControl-style meth-

Table 1: Comparison with controlled-generation methods. The best scores are in bold, and the second scores are under lined.

| Configuration | F1 ↑ / MSE ↓ | FID ↓ | SSIM ↑ | PSNR ↑ | CLIP-T ↑ |
|---|---|---|---|---|---|
| ControlNet SD1.5 (Canny) [47] | 0.23 / * | 18.18 | 0.2585 | 10.55 | 0.3083 |
| ControlNet++ (Canny) [17] | 0.30 / * | 22.06 | 0.2784 | 10.59 | 0.2986 |
| UniControl (Canny) [30] | **0.35** / * | 21.22 | 0.3714 | 11.66 | 0.3103 |
| UniControlNet (Canny) [49] | 0.26 / * | 17.97 | 0.2783 | 10.59 | 0.3137 |
| FLUX.1-dev ControlNet (Canny) [43] | 0.16 / * | 27.11 | 0.2515 | 10.65 | 0.3009 |
| ControlNet SD1.5 (Depth) [47] | * / 30.64 | 18.09 | 0.2383 | 10.22 | 0.3107 |
| FLUX.1-dev ControlNet (Depth) [44] | * / 47.04 | 19.27 | 0.1968 | 10.74 | 0.3087 |
| ControlNet++ (Depth) [17] | * / 27.79 | 23.23 | 0.2093 | 9.71 | 0.3020 |
| UniControl (Depth) [30] | * / 33.51 | 28.24 | 0.2255 | 10.09 | 0.3105 |
| UniControlNet (Depth) [49] | * / 34.72 | 22.25 | 0.2038 | 10.12 | **0.3156** |
| Ours (Iterative Extraction) | 0.30 / **20.76** | 16.43 | **0.8078** | **19.11** | 0.3043 |
| Ours (One Step Extraction) | 0.28 / 21.18 | **15.64** | 0.7564 | 17.49 | 0.3087 |

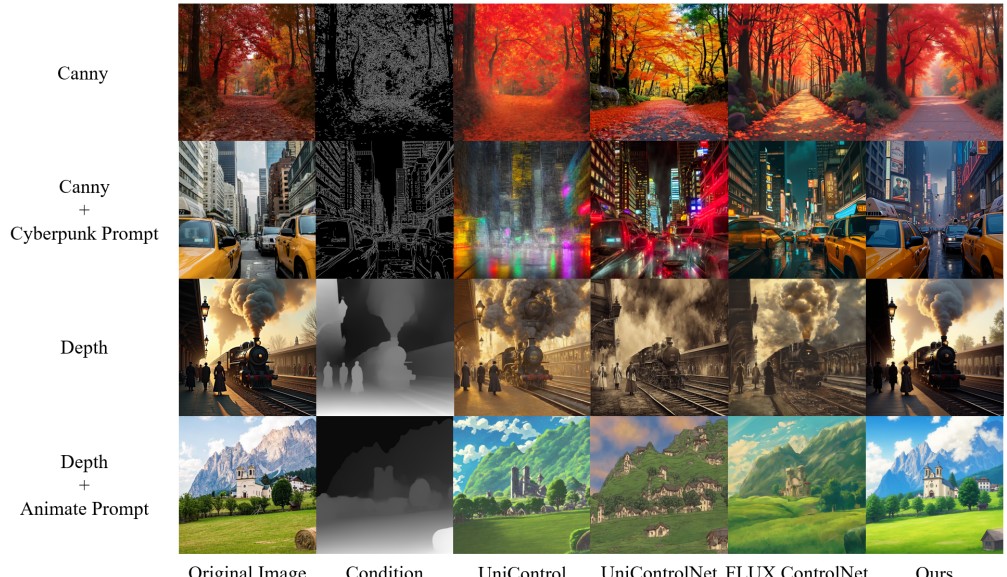

Figure 4: Qualitative comparisons on structure-conditioned image generation. Rows 1 and 3 show results where all methods are conditioned using the original caption of the reference image. Rows 2 and 4 present generations under stylized prompts to evaluate each method's ability to generalize beyond the original content.

ods—which rely on handcrafted edge or depth inputs—our method achieves higher visual fidelity without requiring retraining or specialized condition maps.

In edge-conditioned tasks, FreeControl achieves an F1 score of 0.30 with lower mean squared error (MSE), produces Canny edge results comparable to UniControl and ControlNet++ while using only the raw reference image. Additionally, compared to the iterative extraction, our method performs competitively while exerting significantly less computational resources.

These results demonstrate that FreeControl not only preserves spatial structure and semantic content effectively but also serves as a lightweight, training-free alternative to existing structure-conditioned generation pipelines. For further reference regarding the flexibility and fidelity of our method, we also provide quantitative comparisons that benchmark under different settings in the Appendix.

### 4.3  Qualitative Results

We present qualitative comparisons in fig. 4, where FreeControl is conditioned on raw reference images, while baseline methods rely on preprocessed control signals. Under the original prompt,

FreeControl delivers superior structural alignment and visual fidelity. In contrast, comparison methods either fail to align accurately with the intended structure or generate artifacts such as blur or noise, undermining image quality. We further demonstrate results on representative stylized prompts to evaluate generalization beyond the original setting. FreeControl successfully preserves structural integrity while adapting to new prompts, demonstrating robust guidance under prompt variation. Canny-based methods rigidly adhere to edge maps, often at odds with prompt semantics—resulting in unnatural appearances and ghost artifacts. Depth-based methods suffer from insufficient detail in the control signal, leading to misalignment, semantic drift, and diminished image fidelity. Overall, the results underscore FreeControl's ability to maintain consistent structural control and prompt adherence, even when guided by raw image inputs rather than handcrafted control maps.

## 4.4 Inference Time

We benchmark inference speed for FLUX ControlNet (Canny), FLUX ControlNet (Depth), the vanilla FLUX pipeline, and our method. All models are run with 25 denoising steps and produce $1024 \times 1024$ px outputs on an NVIDIA RTX A6000. For each pipeline, we fix a single prompt and a single source image—together with its corresponding condition map (canny edges or depth)—and execute the generation 100 times, recording the elapsed time at every run. Note that we exclude the pre-process time of the comparison methods for a fair comparison. The aggregate statistics from these 100-run trials are reported in table 2, and our method, being a test-time augmented method, performs equally efficiently as the training-based methods. Beyond that, the additional memory usage brought by our method is around 1 GB, which is also negligible.

Table 2: Inference time of different methods.

| Configuration | Average Inference Time | Max | Min | Variance |
|---|---|---|---|---|
| FLUX Original Pipeline FLUX.1-dev [16] | 24.89 | 24.98 | 24.18 | 0.0101 |
| FLUX.1-dev ControlNet (Canny) [43] | 26.01 | 26.52 | 25.61 | 0.0054 |
| FLUX.1-dev ControlNet (Depth) [44] | 26.01 | 26.09 | 25.80 | 0.0034 |
| Ours | 26.11 | 26.16 | 25.32 | 0.0117 |
| Ours(Iterative Extraction) | 45.16 | 48.09 | 45.01 | 0.1012 |

## 4.5 Compatibility with Fine-Tuned or LoRA-Augmented Models

ControlNet [47] often exhibit limited compatibility with finetuned or augmented via LoRA [9]. This instability arises because ControlNet relies heavily on the original backbone's parameters—its control branches are trained jointly with the base model and assume specific internal feature distributions. However, our method is not dependent on any specific model architecture or weights, demonstrating strong adaptability across different model variants. To validate this, we conduct experiments on both fine-tuned [1] and LoRA-augmented [26] models, comparing our method with ControlNet FLUX (Canny) and FLUX ControlNet (Depth). As shown in fig. 5, our method demonstrates superior compatibility by providing stable results with consistent structure and fine adaptation to the model changes in the community models, whereas ControlNet fails to be compatible with them and produce artifacts and distorted results. More results can be found in the Appendix.

## 5 Ablation Study

### 5.1 One-Step vs. Iterative Attention Extraction

To validate the effectiveness of extracting attention from a single timestep, we compare our method against a baseline that mimics iterative attention extraction across multiple denoising steps—similar to inversion-based or reconstruction-based strategies. In this baseline, attention matrices are extracted and injected step-by-step, rather than reused. As shown in table 1 and table 2, one-step injection achieves comparable structural fidelity while significantly reducing computational overhead. This result supports our hypothesis that structural information can be captured once and reused without loss of guidance, due to the shared purpose of structural encoding across timesteps.

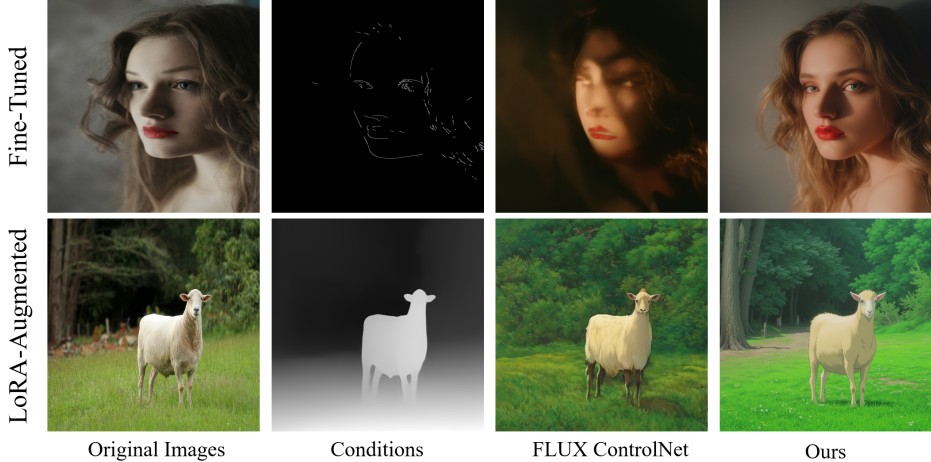

Figure 5: Examples of compatibility with fine-tuned or LoRA-augmented models.

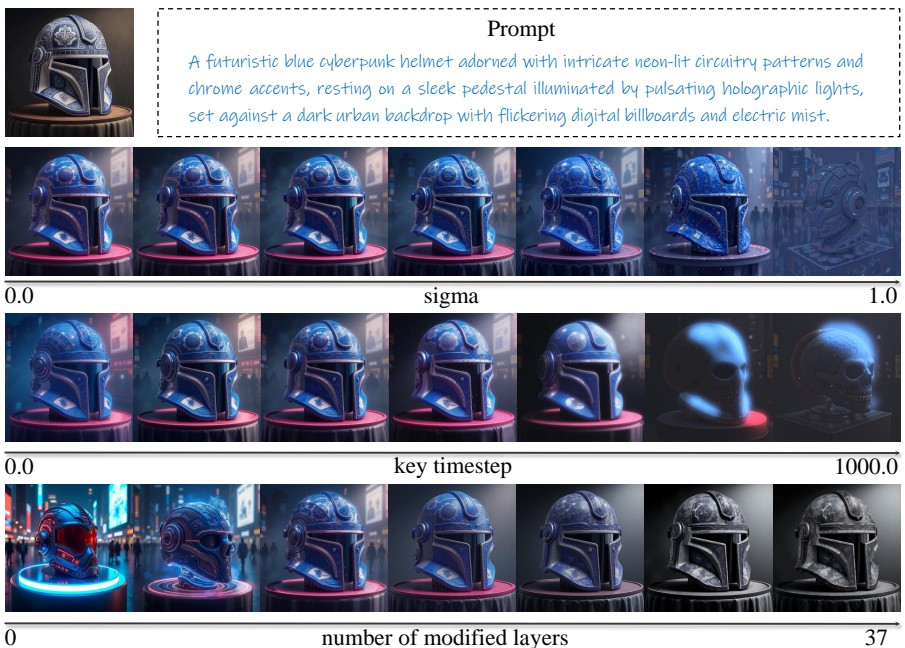

Figure 6: Visual analysis of structural control effects by varying injection depth, sigma, and key timestep in FreeControl.

## 5.2 Injection Depth vs. Sigma vs. Key timestep

The injection depth (number of transformer layers) influences the strength of structural control, while the injection quality (i.e., the content of the attention matrix) determines its focus. To isolate the effects of each factor, we vary it individually while holding others fixed at empirically optimal values. As shown in fig. 6, varying the injection depth reveals a different trade-off: injecting into fewer transformer layers relaxes structure in less critical regions, improving texture and color fidelity, whereas deeper injection increases rigidity at the cost of visual richness—especially in color saturation. The choice of key timestep, on the other hand, influences the focus of attention—that is, what kind of structural information is being injected. Earlier timesteps (e.g., $t = 0$) yield more abstract, layout-level attention that allows greater freedom in fine details; later timesteps (e.g., $t = 661$) still capture high-level structure but with greater specificity and finer granularity, resulting in more detailed structural alignment. The sigma value, in contrast, remains relatively stable at moderate settings, with

Prompt
A towering lighthouse blazes atop a jagged crimson cliff, surrounded by a fiery red sky and storm clouds. A winding stone path glows like molten cracks, while a blood-red ocean crashes below. Dark towers flank the scene, silhouetted against the burning sky. The world is an intense, surreal inferno of passion, violence, and awe, with volcanic energy and dramatic dystopian style.

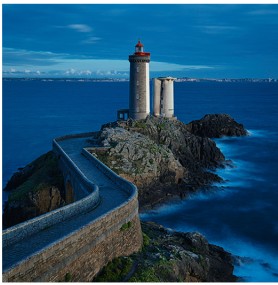 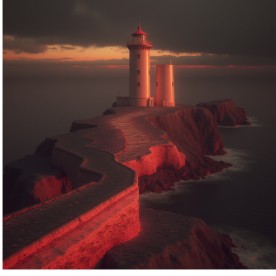 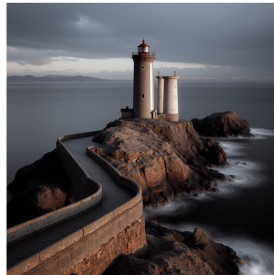

Original Image        Layer-aware Injection        Full-layer Injection

Figure 7: Qualitative Comparison between Layer-Aware Injection and Full-Layer Injection

structural control gradually diminishing as it approaches 1—first affecting fine details, then larger structures. Based on these observations, we recommend adjusting layer depth and injection range to tune structural strength, while selecting the appropriate key timestep and sigma to steer the level of structural detail captured in the generation.

### 5.3 Impact of Layer-Aware Injection

As discussed in section 3.1, mid-to-late layer queries encode structural layout rather than raw appearance, aligning better with our structural control objective. However, early-layer queries primarily encode low-level appearance statistics, especially color. When injected, they conflict with the prompt-conditioned key/value features, which often imply a different color palette. This feature mismatch causes the model to lose chromatic fidelity, leading to muted or grayscale-like outputs. To verify this, we test prompts with deliberately shifted color themes. As shown in fig. 7, full-layer injection causes localized color fading in regions where prompt colors diverge from the reference, while our layer-aware injection maintains both structure and visual quality.

## 6 Limitations

While FreeControl is efficient, training-free, and offers strong structural control, it does not support condition maps like edges or segmentation. This limits scenarios where users prefer editing sketches or symbolic inputs. Although compositional image assembly provides flexibility, some use cases may still benefit from explicit support for sparse conditions.

## 7 Conclusion

This paper revisits a central assumption in attention-based structural control for diffusion models: that effective guidance requires multi-step extraction. We show that a single-step extraction—when properly conditioned—can offer strong, reusable structural signals without inversion or retraining. Our Latent-Condition Decoupling (LCD) reveals that attention quality depends not just on the timestep, but on how the noised latent and conditioning signal are configured. This enables more stable and controllable generation. Beyond efficiency, FreeControl supports intuitive control by allowing users to compose reference images that express both layout and intent—bridging structure and semantics without relying on edge maps or segmentation masks. Overall, our findings suggest attention can serve not just as an internal mechanism, but as a practical, tunable approach for control diffusion models.

# 8 Acknowledgments

This work was supported by the National Natural Science Foundation of China (Grant No. 62406134), Jiangsu Provincial Science & Technology Major Project (Grant No. BG2024042), the Suzhou Key Technologies Project (Grant No. SYG2024136) and the Nanjing University-China Mobile Communications Group Co. Ltd. Joint Institute.

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

# A   Quantitative Results on Stylized Prompts (COCO Dataset)

FreeControl conditions generation directly on the raw image, rather than on derived conditions like edges or depth. This design provides a rich and detailed structural prior, which has proven effective in preserving the original layout and composition during generation. While prior experiments focus on tasks where the target image shares semantic similarity with the reference, real-world use cases often involve prompts that diverge stylistically or conceptually from the original image. To broaden the evaluation scope and assess FreeControl's robustness in more diverse generative settings, we conduct a benchmark using stylized prompts.

We construct this benchmark on the COCO [20] validation set. For each image, we retain the original as the structural reference and generate stylized prompts by combining the original caption with one of five target styles (e.g., Cyberpunk, Vaporwave). These stylized prompts are created using GPT-4o [25], conditioned on both the caption and the style keyword.

This setup introduces a challenging mismatch between the style and the visual structure, making it difficult for models to retain both prompt adherence and structural fidelity. We compare FreeControl against Flux-ControlNet [47], which serves as a strong baseline for structure-conditioned generation on the same backbone. As shown in table 3, FreeControl consistently preserves structure better (as measured by F1 and MSE) while generating content more semantically aligned with the stylized prompts (via CLIP-T). This demonstrates the strength of our method in transferring structure faithfully even when prompt semantics diverge from the original image.

# B   More Results on Compatibility with Fine-Tuned or LoRA-Augmented Models

To further support the findings discussed in the main paper, we provide additional qualitative results on fine-tuned and LoRA-augmented diffusion models. Specifically, we evaluate FreeControl and FLUX ControlNet variants on community models that are fine-tuned [1, 11] or LoRA-augmented [26, 29].

As shown in figs. 8 to 11, our method consistently preserves structure and semantic fidelity across diverse model variants, producing stable and visually coherent outputs. In contrast, ControlNet-based approaches exhibit visible artifacts, color shifts, or loss of structural alignment under the same settings.

These results further confirm that FreeControl maintains strong compatibility across both fine-tuned and LoRA-augmented backbones, benefiting from its training-free nature and independence from specific model weights or feature distributions.

# C   Additional Visual Results

We provide more visual results for the readers to reference. fig. 12 showcases additional generation results based on compositional reference images. Users can crop and paste objects into a layout to specify spatial intent, allowing for precise control over the scene's composition. With a touch of creativity, FreeControl empowers users to bring their imaginative visions to life, generating stunning, dynamic visuals that reflect their unique concepts (like a giant floating whale in the sky). fig. 13 and fig. 14 provide further comparisons between FreeControl and FLUX ControlNet.

Table 3: Quantitative evaluation on the COCO validation set using stylized prompts. The best scores are in bold.

| Method | F1 ↑ | MSE ↓ | SSIM ↑ | PSNR ↑ | CLIP-T ↑ |
|---|---|---|---|---|---|
| FLUX ControlNet (Canny) | 0.19 | N/A | 0.2629 | 9.72 | 0.2646 |
| FLUX ControlNet (Depth) | N/A | 41.41 | 0.2029 | 9.98 | 0.2448 |
| Ours | **0.25** | **26.24** | **0.4825** | **15.07** | **0.2981** |

# D  Expanded Baseline Comparison Results

We have added quantitative comparison experiments with two image editing models, In-Context Edit [48] and Taming Rectified Flow [42], to improve the fairness and completeness of the evaluation. The test images are consistent with the previous comparison experiments. We use image captions and stylized prompts as text inputs respectively, and the corresponding results are shown in table 4 and table 5.

According to the results, Our method performs comparably to, and often surpasses, Taming Rectified Flow across several metrics. ICEdit, built on the FLUX-Fill model [16] for image inpainting, achieves relatively high similarity metrics (e.g., PSNR) primarily because it keeps all content outside the edited region untouched. However, this strategy limits its ability to satisfy the desired balance between structural control and free content generation. As a result, its CLIP-T score is lower, and it often struggles with stability and controllability when following editing instructions.

Table 4: Quantitative Results with image-editing methods on the COCO validation subset.

| Method | F1 ↑ | MSE ↓ | SSIM ↑ | PSNR ↑ | CLIP-T ↑ | FID ↓ |
|---|---|---|---|---|---|---|
| In-Context Edit | 0.47 | 17.30 | 0.7781 | 21.34 | 0.3024 | 8.64 |
| Taming Rectified Flow | 0.19 | 28.31 | 0.4390 | 16.90 | 0.3164 | 16.35 |
| Ours | 0.28 | 21.18 | 0.7564 | 17.49 | 0.3087 | 15.64 |

Table 5: Quantitative Results with image-editing methods on the COCO validation subset using stylized prompts.

| Method | F1 ↑ | MSE ↓ | SSIM ↑ | PSNR ↑ | CLIP-T ↑ |
|---|---|---|---|---|---|
| In-Context Edit | 0.30 | 35.96 | 0.5436 | 14.70 | 0.2543 |
| Taming Rectified Flow | 0.17 | 38.49 | 0.4034 | 16.37 | 0.2585 |
| Ours | 0.25 | 26.24 | 0.4825 | 15.07 | 0.2981 |

# E  Additional Quantitative Ablation Analysis

We have supplemented more comprehensive ablation studies to justify the choice of key parameters in our method, such as the key timestep $t^*$, numbers of modified transformer layers and $\sigma$. The results are presented in table 6. Thanks to LCD, by flexibly tuning the hyperparameters, we can achieve structural control of varying strength and granularity, producing stable and controllable results that cater to different user requirements.

Table 6: Ablation results on on the COCO validation subset. Each entry in the Parameters column indicates the number of modified layers, the key timestep $t^*$, and the $\sigma$, in that order.

| Parameters | F1 ↑ | MSE ↓ | SSIM ↑ | PSNR ↑ | CLIP-T ↑ | FID ↓ |
|---|---|---|---|---|---|---|
| 20-661-0.25 | 0.27 | 23.01 | 0.5251 | 16.61 | 0.3083 | 17.84 |
| 25-561-0.25 | 0.27 | 22.44 | 0.5232 | 16.61 | 0.3077 | 17.42 |
| 25-661-0.0 | 0.30 | 21.74 | 0.5630 | 17.20 | 0.3061 | 17.78 |
| 25-661-0.25 | 0.28 | 21.86 | 0.5438 | 16.87 | 0.3048 | 18.00 |
| 25-661-0.5 | 0.24 | 24.64 | 0.5097 | 16.54 | 0.3072 | 22.51 |
| 25-761-0.25 | 0.28 | 23.47 | 0.5492 | 17.07 | 0.3048 | 21.40 |
| 30-661-0.25 | 0.30 | 22.20 | 0.5461 | 16.88 | 0.3020 | 20.20 |

# F  Evaluation under Challenging Structural Scenarios

## F.1  Semantic Entanglement and Object Occlusion

We conduct stress tests on both real and synthetic images featuring semantic entanglement and severe object overlap. Selected results are shown in fig. 15. For each example, the left image is the original input, and the right image is the generated result. As observed, FreeControl consistently avoids distorted or unrealistic artifacts such as extra limbs or warped body structures. The outputs remain natural-looking and visually coherent, demonstrating strong robustness even under highly challenging compositional scenarios.

## F.2  Preservation of Facial Identity

Our method provides flexible control over facial identity preservation, allowing users to adjust the strength of identity retention via hyperparameters. Under non-conflicting text-image guidance, tuning parameters such as the number of modified transformer layers enables strong structural control while preserving facial details, making FreeControl suitable for identity-sensitive tasks. As shown in fig. 16, with higher control strength, our approach demonstrates a strong ability to retain facial identity. Conversely, for artistic creation or diversity-oriented generation, relaxing the control allows for slight, intentional changes in facial features, leading to more expressive results.

# G  Applicability to UNet-based Models (i.e., Stable Diffusion)

Our method is designed to operate purely at the attention level, making it architecture-agnostic. We have implemented it on UNet–based models (e.g., SD1.5 [33], SDXL [28]) and observed strong structural control behavior after only minimal hyperparameters adjustments to fit the model. Several qualitative examples of structural control are presented in fig. 17 and fig. 18. We further note that, due to the inherent capacity limitations of UNet-based models, the degree of controllability can diminish in highly complex scenarios. In practice, we find that leveraging FLUX models [16] yields more stable and visually coherent generations, and we recommend their use when high-fidelity control is desired.

# H  Further Discussion on the Design Space

## H.1  Independence from ROPE

The query matrices FreeControl extracts are captured before RoPE [39] is applied, so the injected queries contain no positional encoding — they are entirely image-driven. While RoPE still affects key and value during generation, it does not alter what FreeControl injects. Furthermore, FreeControl works identically on U-Net architectures (which do not use RoPE), showing that structural consistency stems from the extracted queries themselves, not from positional priors.

To directly confirm this point, we ran a controlled test by removing RoPE entirely from the FLUX model. As expected, the base model collapsed into near-random noise, since it was never trained to operate without positional encoding. Crucially, when we applied FreeControl under the same no-RoPE setup, the one-step injection still imposed clear, image-driven structure on the output. The result looked like "structured noise" faithfully echoing the condition image's layout — strong evidence that FreeControl's guidance originates from the injected queries themselves, not from RoPE.

## H.2  Key timestep Choice

The key timestep fundamentally governs the granularity of structural information that FreeControl can extract. In diffusion models, each denoising step is influenced not only by progressively refined latents but also by a changing timestep input that biases the network toward different levels of detail. Conceptually, the key timestep acts like a focus knob: adjusting it continuously shifts the model's representational emphasis from global layout patterns to fine-grained textures.

By holding the latent fixed and sweeping only the key timestep, our experiments reveal a natural progression in structural granularity encoded within the query matrices. Both quantitative metrics and

visual evidence show a smooth evolution—from coarse shape-level representation toward detailed texture-level encoding. Among all tested values, key timestep (661) emerges as a sweet spot, offering the best trade-off between global consistency and structural precision, making it the most suitable extraction point for query-based control.

### H.3 Layer-wise Query Matrices Similarity

We extract query matrices at different depths under two configurations: with LCD and without LCD, where the only difference lies in whether noise is added to $x_0$ during the forward diffusion process (as defined in Sec. 3.1 of main paper). We then compare these two sets of query matrices with the query matrices from every timestep of the multi-step extraction variant and compute the cosine similarity, as reported in table 7. The layer-wise similarity shows a clear low-to-high trend: shallow layers tend to have lower similarity, while deeper layers converge more. This aligns with our layer-aware injection choice in Sec. 3.1 — early layers focus more on appearance elements rather than structure, diverging more across timesteps and contributing less to shared structural signals.

Table 7: Cosine Similarity Between One-Step and Multi-Step Extracted Query Matrices. The "Global" row reports the average similarity of query matrices across all layers.

| Layer Depth | Cosine Similarity | |
| --- | --- | --- |
| | w/ LCD | w/o LCD |
| Early | 0.5418 | 0.6680 |
| Mid | 0.5861 | 0.6853 |
| Last | 0.6011 | 0.7638 |
| Global | 0.5769 | 0.7063 |

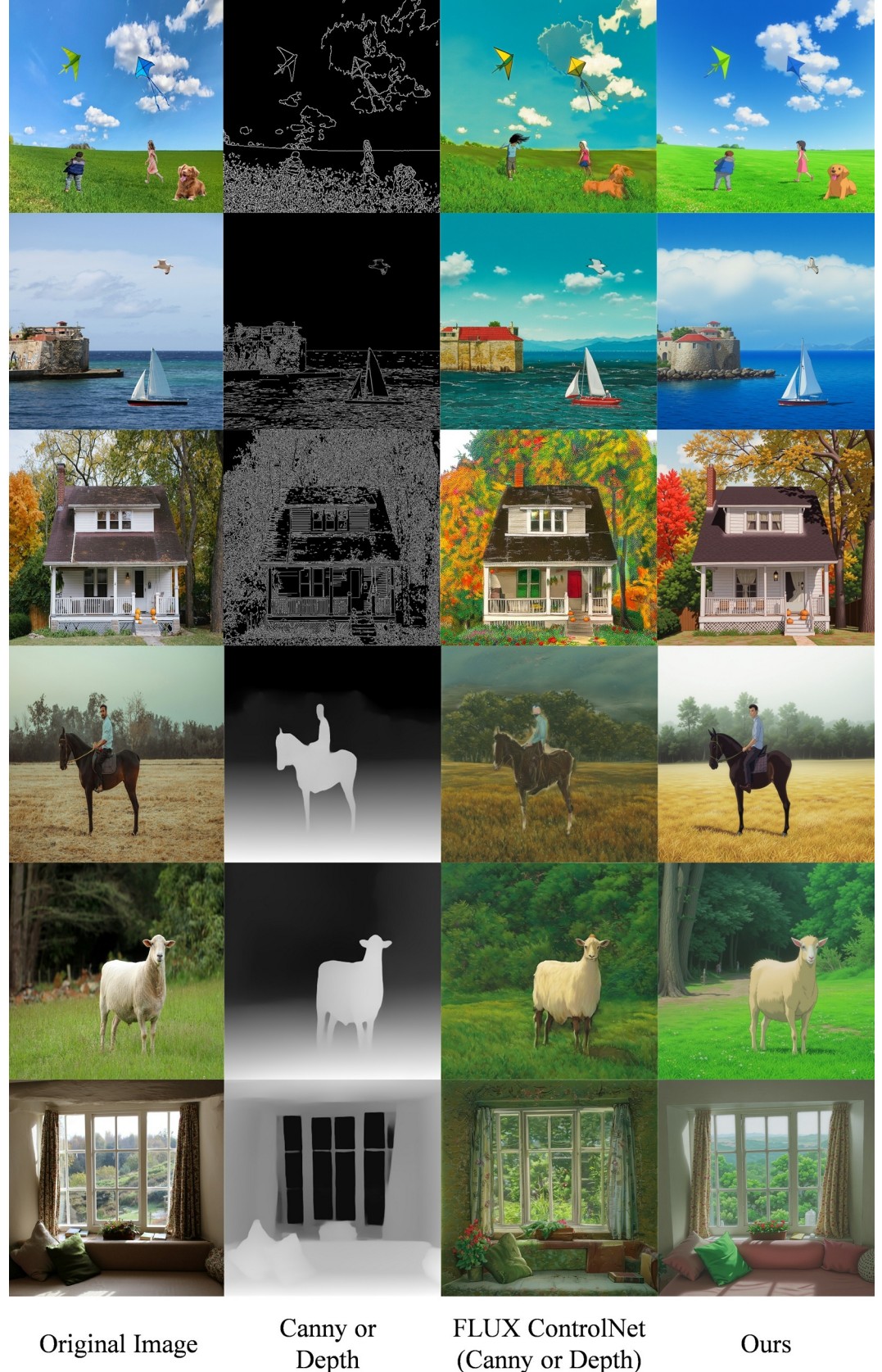

| Original Image | Canny or Depth | FLUX ControlNet (Canny or Depth) | Ours |

Figure 8: Visual results comparing our method and FLUX ControlNet on Lora-Augmented models (Ghibli-Style LoRA).

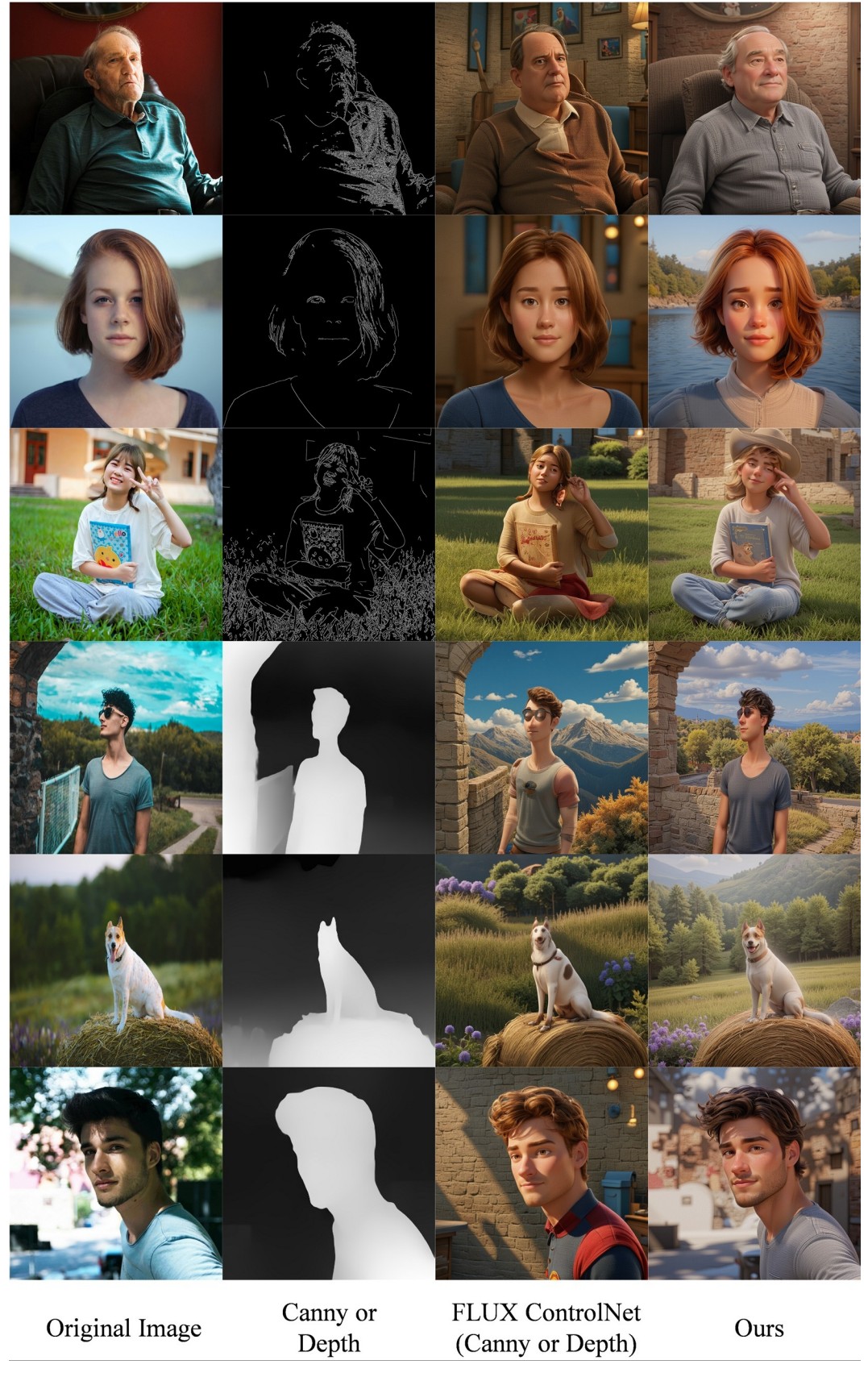

| Original Image | Canny or Depth | FLUX ControlNet (Canny or Depth) | Ours |

Figure 9: Visual results comparing our method and FLUX ControlNet on Lora-Augmented models (Canopus-Pixar-3D-Style LoRA).

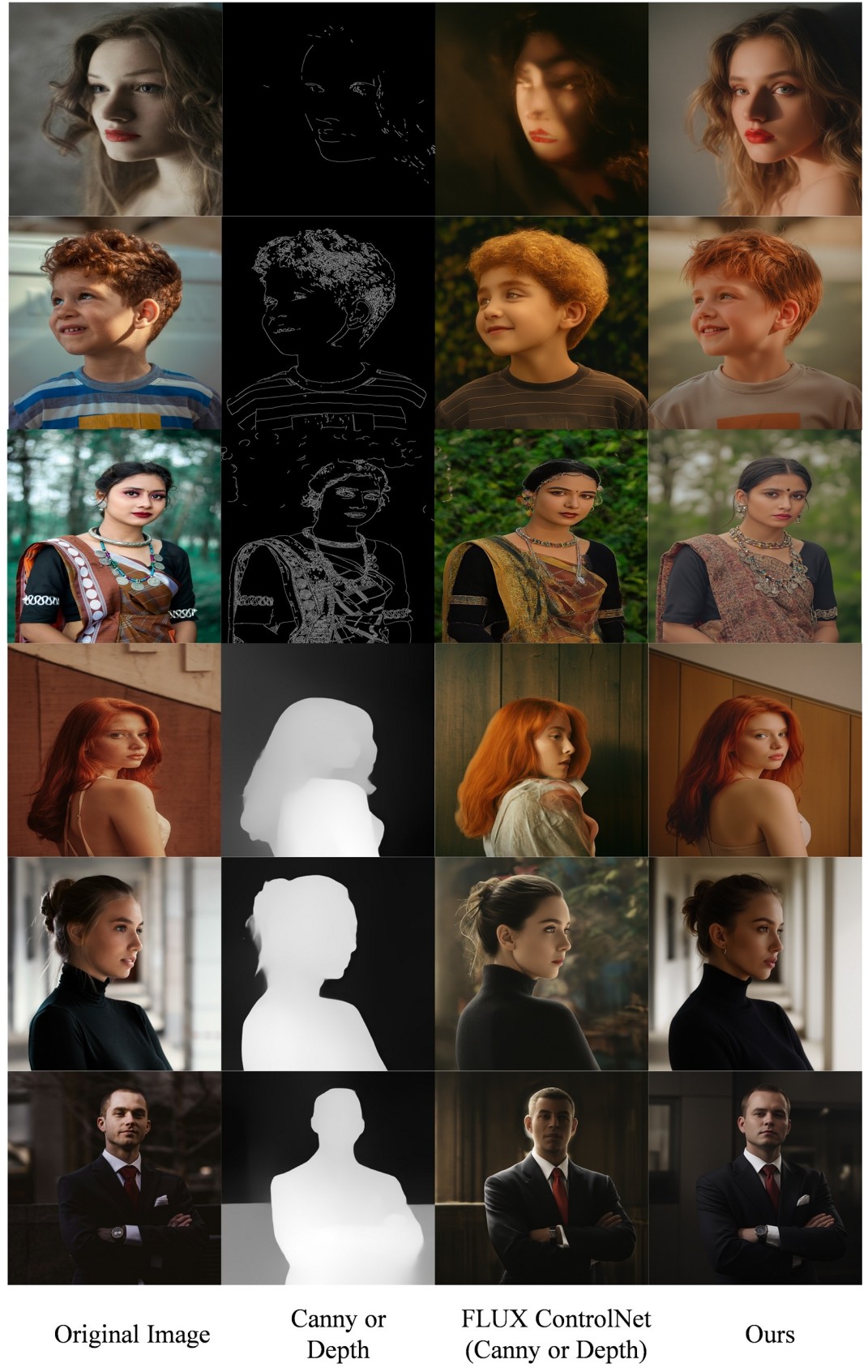

Original Image        Canny or Depth        FLUX ControlNet (Canny or Depth)        Ours

Figure 10: Visual results comparing our method and FLUX ControlNet on finetuned models (AWPortrait Fine-Tune).

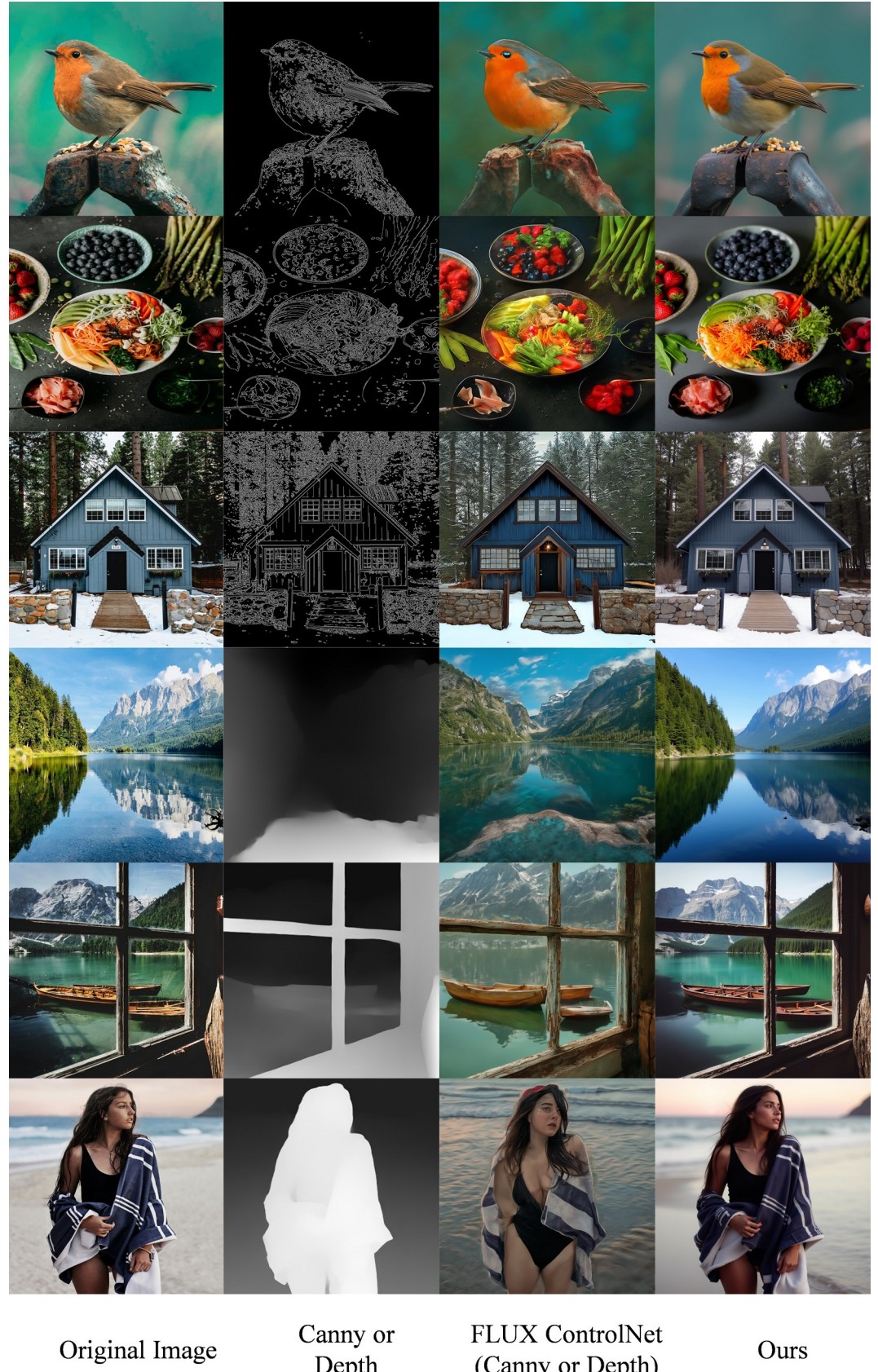

| Original Image | Canny or Depth | FLUX ControlNet (Canny or Depth) | Ours |

Figure 11: Visual results comparing our method and FLUX ControlNet on finetuned models (UltraReal Fine-Tune).

*Compose Content*

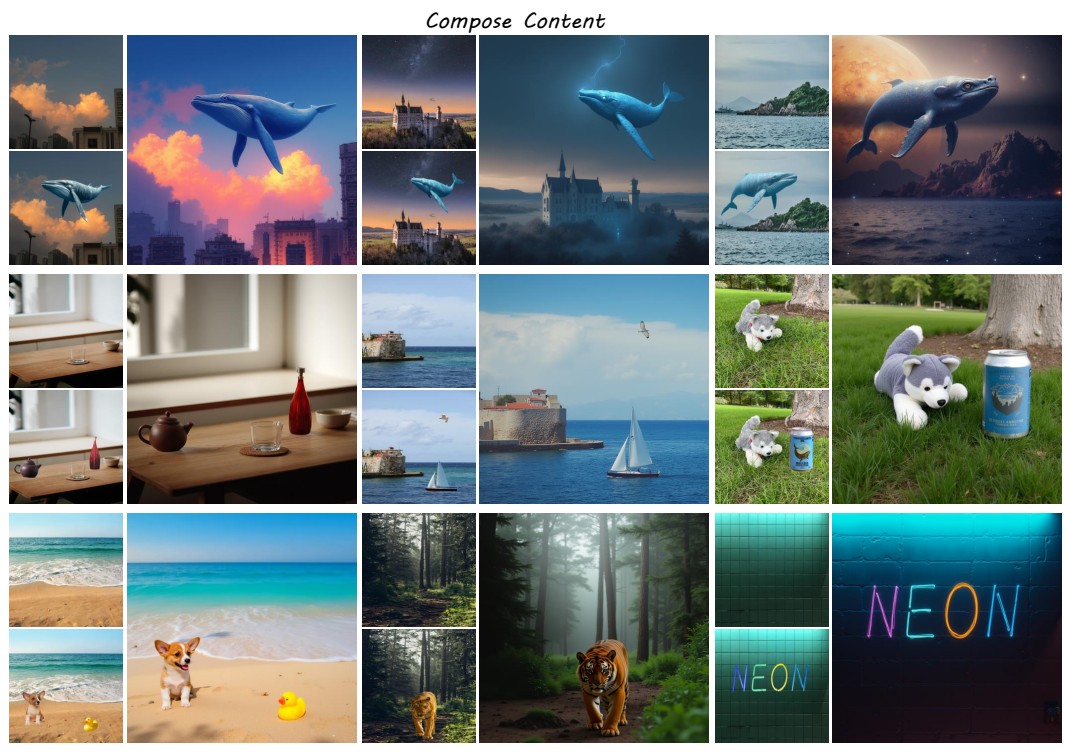

Figure 12: More visual results on compositional generation.

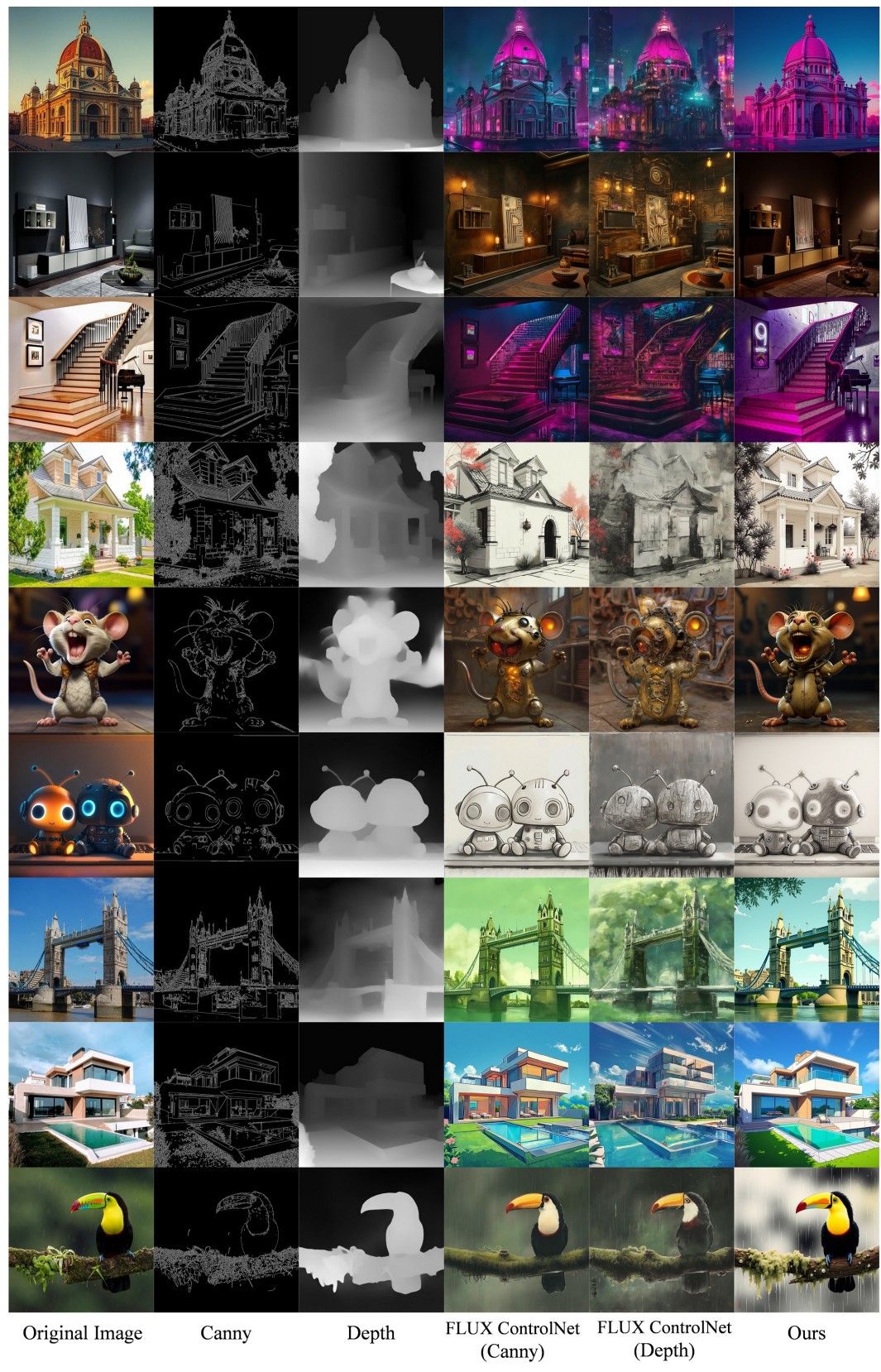

|   |   |   |   |   |   |
|---|---|---|---|---|---|
| Original Image | Canny | Depth | FLUX ControlNet (Canny) | FLUX ControlNet (Depth) | Ours |

Figure 13: Qualitative comparisons on structure-conditioned image generation.

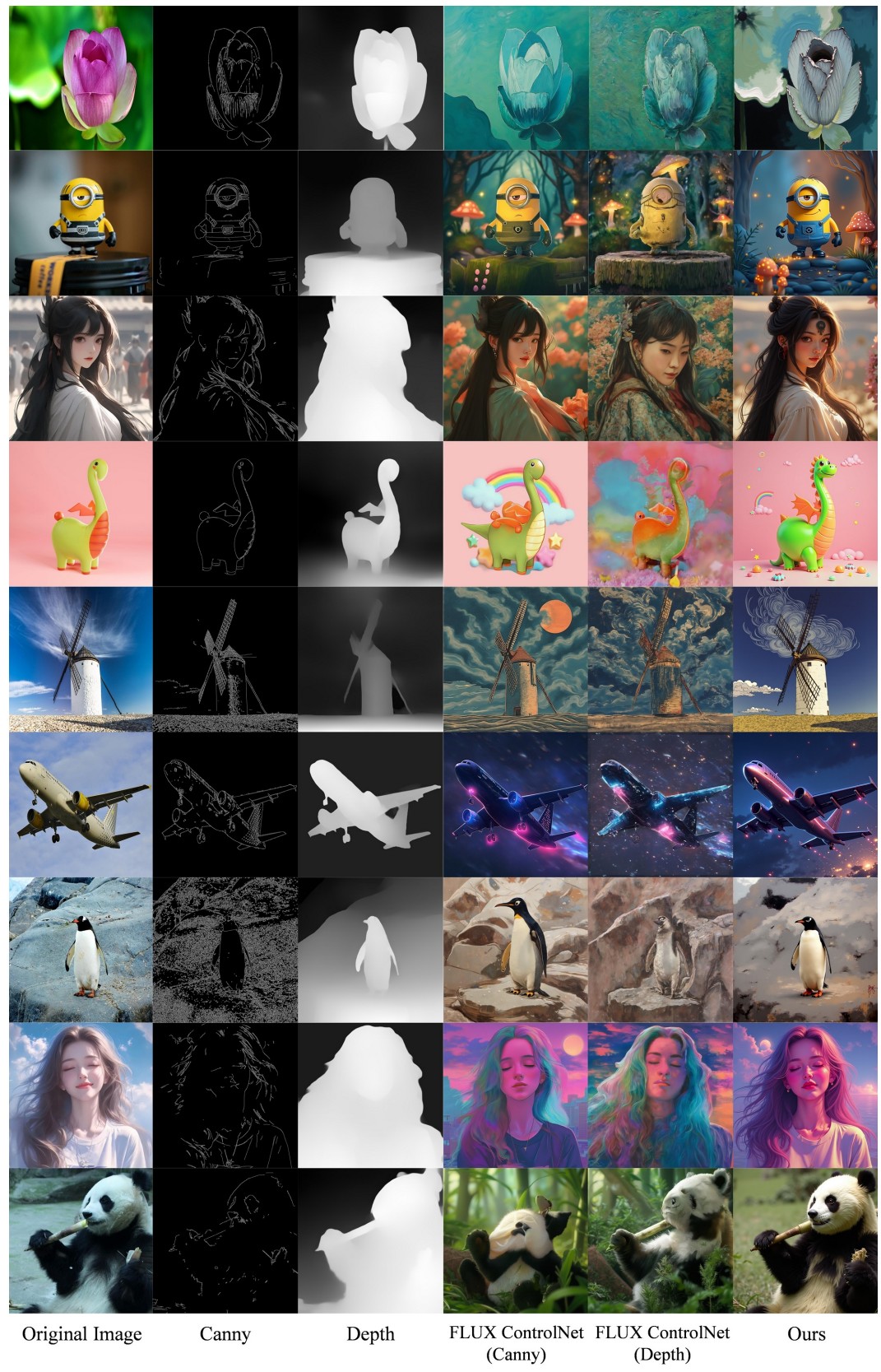

Original Image     Canny     Depth     FLUX ControlNet (Canny)     FLUX ControlNet (Depth)     Ours

Figure 14: Qualitative comparisons on structure-conditioned image generation.

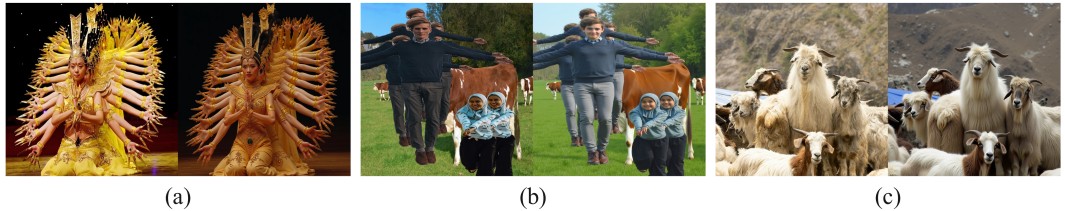

Figure 15: Examples of generated images under Semantic Entanglement and Object Occlusion. For each pair, the image on the left is the original image, and the image on the right is the generated result.

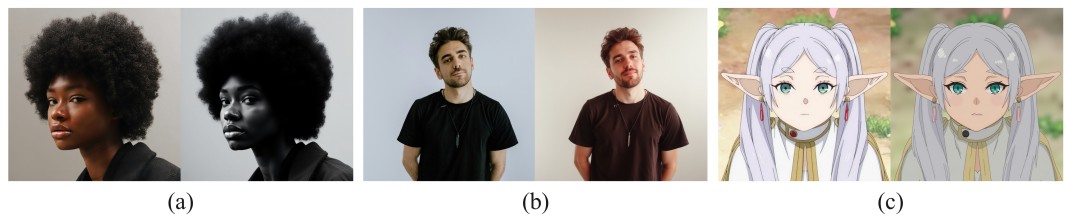

Figure 16: Examples of Facial Identity Control. Adjusting control strength, together with a suitable prompt, enables strong structural preservation of facial features. For each pair, the image on the left is the original image, and the image on the right is the generated result.

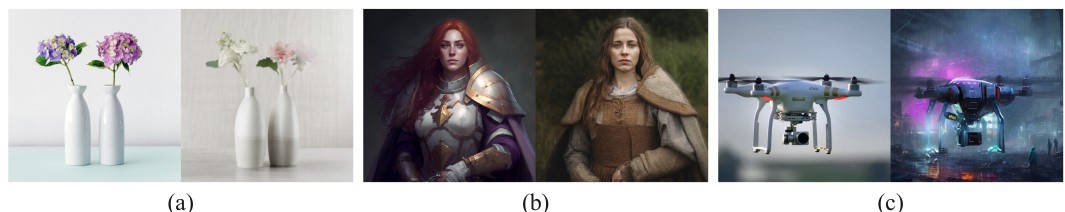

Figure 17: Generation Examples on SDXL model Using Our Method. For each pair, the image on the left is the original image, and the image on the right is the generated result.

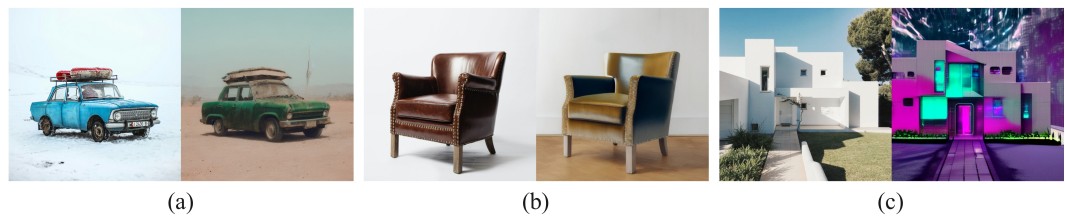

Figure 18: Generation Examples on SD-1.5 model Using Our Method. For each pair, the image on the left is the original image, and the image on the right is the generated result.

