# OpenReview forum: "FreeControl: Efficient, Training-Free Structural Control via One-Step Attention Extraction"
_NeurIPS.cc/2025/Conference — NeurIPS 2025 poster_

### Official Review · Reviewer_RWpJ · 2025-06-20

**Clarity:** 3
**Significance:** 3
**Originality:** 3
**Rating:** 5
**Confidence:** 5

**Summary:**

This paper proposes a new training-free strategy for structural control by using only one-step attention extraction. Specifically, the method extracts the attention query matrix from a specific time step and certain layers during the inversion process and injects it into each time step during denoising. This approach greatly simplifies the process and offers a novel direction for future research. Experimental results demonstrate the effectiveness of the proposed method.

**Questions:**

Please try to solve the questions proposed in weaknesses section.

**Ethical Concerns:**

["NO or VERY MINOR ethics concerns only"]

**Final Justification:**

The authors have provided detailed and thoughtful responses during the rebuttal, which have largely addressed my initial concerns. I appreciate the additional experiments and clarifications, and I believe the planned inclusion of the qualitative and quantitative comparisons discussed in the rebuttal will further improve the clarity and persuasiveness of the paper. Overall, I support acceptance.

**Limitations:**

yes

**Quality:**

3

**Strengths And Weaknesses:**

## Strengths
1. The proposed method is simple yet effective. By extracting attention features from only a single step, it significantly reduces the operational complexity compared to previous approaches.
2. The presented results are impressive. Moreover, the method demonstrates natural and coherent composition when combining multiple contents.
3. The overall writing is clear and easy to follow, making the paper accessible and understandable.

## Weaknesses
- The paper lacks sufficient quantitative and comprehensive experimental analysis to justify the selection of specific time steps and layers (e.g., time step 661 used in the paper). Relying solely on empirical evaluation seems somewhat arbitrary and raises questions about whether better choices of time steps and layers might exist. Moreover, if the underlying model (e.g., FLUX) were replaced with SD3 or another DiT-based model, would the same choices still apply? Are the observed patterns model-specific or generalizable? A more rigorous analysis—similar to what was provided in **Stable Flow: Vital Layers for Training-Free Image Editing [CVPR 2025]**—would help to support the design decisions and strengthen the method's justification.
- The comparison baselines used in the paper are mostly outdated models, which weakens the credibility of the experimental results. Since the proposed method is built upon a strong prior model (Flux .1-dev based on DiT), the improved performance may partially result from the strength of the underlying model itself rather than the proposed strategy. It would be more convincing to include comparisons with recent SOTA models that are also based on Flux or DiT architectures. The authors are encouraged to consider adding comparisons with the following works:
    - **Taming Rectified Flow for Inversion and Editing**
    - **OminiControl: Minimal and Universal Control for Diffusion Transformer**
    - **EasyControl: Adding Efficient and Flexible Control for Diffusion Transformer**
    - **In-Context Edit: Enabling Instructional Image Editing with In-Context Generation in Large Scale Diffusion Transformer**

    Alternatively, adapting the proposed method to smaller-scale DiT models such as SD3 could help demonstrate its robustness and generalizability.

- In the examples involving human generation and editing, it appears that facial identity is not well preserved. This suggests that the method's controllability may still be limited, especially in more fine-grained or identity-sensitive tasks.

---

> ### Author Rebuttal · Authors · 2025-07-29
>
> ### Rebuttal to Weakness 1: Quantitative Analysis of Ablations
>
> We appreciate Reviewer 4’s call for deeper quantitative analysis of the ablations — this is a fair point and will strengthen the paper. Our initial focus on **qualitative illustrations** in Sec. 5.2 and Fig. 5 was driven by two considerations:
>
> 1. The shifts caused by hyperparameters (layer depth, σ, timestep condition) produce **very visible changes** — such as structure tightening or relaxing and granularity change — rather than subtle metric tweaks. We believed showing these striking differences directly would help readers intuitively understand how each setting affects the result.
>
> 2. Because these parameters can combine in many ways, exhaustive metrics for every combination would create a prohibitively large experiment grid.
>
> That said, we fully agree that numerical backing is important. **We have already run a minibatch of quantitative ablations** and include those results below, with more comprehensive runs planned for the camera‑ready.
>
> #### Table A: Mini‑Batch Quantitative Ablation Results
>
> | Layers | Timestep | Sigma | F1 ↑ | MSE ↓ | SSIM ↑ | PSNR ↑ | CLIP-T ↑ | FID ↓ |
> |--------|----------|-------|------|--------|--------|--------|-----------|--------|
> | 25     | 561      | 0.25  | 0.27 | 22.60  | 0.7504 | 17.42  | 0.3072    | 71.40  |
> | 25     | 761      | 0.25  | 0.28 | 23.89  | 0.7727 | 17.95  | 0.3042    | 78.03  |
> | 25     | 661      | 0.00  | 0.29 | 22.15  | 0.7874 | 18.10  | 0.3056    | 69.94  |
> | 25     | 661      | 0.50  | 0.24 | 21.94  | 0.7338 | 17.46  | 0.3041    | 79.13  |
> | 30     | 661      | 0.25  | 0.30 | 22.79  | 0.7835 | 17.82  | 0.3014    | 75.58  |
> | 25     | 661      | 0.25  | 0.28 | 21.89  | 0.7681 | 17.70  | 0.3062    | 70.70  |
>
> #### Table B: Mini‑Batch Quantitative Ablation Results on Stylized COCO
>
> | Layers | Timestep | Sigma | F1 ↑ | MSE ↓ | SSIM ↑ | PSNR ↑ | CLIP-T ↑ |
> |--------|----------|-------|------|--------|--------|--------|-----------|
> | 25     | 561      | 0.25  | 0.22 | 27.80  | 0.6910 | 15.66  | 0.2425    |
> | 25     | 761      | 0.25  | 0.25 | 26.92  | 0.7485 | 17.03  | 0.2411    |
> | 25     | 661      | 0.00  | 0.26 | 25.20  | 0.7589 | 17.20  | 0.2606    |
> | 25     | 661      | 0.50  | 0.19 | 25.87  | 0.6801 | 15.62  | 0.2396    |
> | 25     | 661      | 0.25  | 0.25 | 30.80  | 0.7258 | 16.30  | 0.2427    |
> | 30     | 661      | 0.25  | 0.28 | 24.08  | 0.7728 | 17.28  | 0.2404    |
>
>
>
> On **timestep 661**: the discussion in **L166** and the sweep in **Sec. 5.2/Fig. 5** already show that we evaluated the entire range, and 661 provided the best balance for Flux. There is also a clear **trend from both directions** — from timestep 0 moving up toward 661 *and* from timestep 1000 moving down toward 661 — where **structural alignment and generation quality steadily improve, peaking around the optimal region near 661**. The partial quantitative results are reflected in Table A, and a more complete version will be provided later.
>
> When we tested FreeControl on other backbones, the exact numbers shifted — as expected — but the **principles for selecting these hyperparameters and how they affect control remained consistent**. These principles are carefully embedded into the **method section** so they can guide future applications of FreeControl to new models.
>
> We thank the reviewer again for highlighting this — it prompted us to expand our quantitative experiments and commit to adding **more rigorous analysis from multiple dimensions** in the paper and appendix to fully justify these design choices. An additional analysis regarding matrix similarity is also provided in our response to Reviewer T9YS, which may be referenced for further insights.
>
>
> ---
>
> ### Rebuttal to Weakness 2: Baselines
> We sincerely thank Reviewer 4 for emphasizing the importance of comparing FreeControl against recent Flux/DiT‑based control methods and for providing helpful recommendations. **This was a valuable point, and we acted on it directly.**
>
> Using released code, we ran **In‑Context Edit** and **Taming Rectified Flow**, adding their results in **Table A** below. The other suggested methods currently have no public implementations, but we will include them in the camera‑ready version once code becomes available.
>
> As shown in **Table A**, FreeControl performs comparably to — and in several cases surpasses — Taming Rectified Flow across key metrics. **In‑Context Edit (ICEdit)**, built on the FLUX.Fill model for **image completion (filling areas in existing images)**, scores strongly on similarity metrics because its base model is highly conservative in retaining existing content, but is performing less optimally on CLIP-T on both Table A and B. Visual results from the experiments in Table B further highlight its limitations: despite favorable metrics, ICEdit frequently produces unnatural and low-quality outputs that appear unrealistic or visually jarring—such as disjointed objects, missing body parts, or incoherent scenes. In contrast, our method demonstrates significantly greater stability and consistently generates higher-quality, more natural, and human-acceptable results
>
> We chose **Flux** intentionally to demonstrate FreeControl’s effect on a strong, modern backbone — but stronger generative quality alone does not guarantee easier structural control. That is why we prioritized adding **new baselines built on Flux itself**. These comparisons show that our method performs comparably or better than other Flux‑based methods on metrics, and that those methods often produce generations with severe visual flaws. In contrast, FreeControl consistently produces stable, natural‑looking images, which is essential for applicability.
>
> We also note that **Table 1** of the paper already includes **ControlNet‑Flux**, the latest Flux/DiT ControlNet adaptation, ensuring FreeControl is  also evaluated against modern Flux‑based pipelines.
>
>
>
> #### Table A: Mini‑Batch Expanded Baseline Comparison Results
>
> | Method                 |  F1 ↑ | MSE  ↓  | SSIM ↑  | PSNR ↑ | CLIP-T ↑ | FID ↓ |
> |-----------------------|------|-------|----------|--------|-----------|--------|
> | In‑Context Edit       |  0.43 | 24.63 | 0.8050   | 19.68  | 0.2787    | 67.63  |
> | Taming Rectified Flow |  0.19 | 28.22 | 0.6131   | 17.41  | 0.3160    | 72.49  |
> | FreeControl           |  0.28 | 21.21 | 0.7580   | 17.43  |0.3076       | 66.72  |
>
> #### Table B: Mini‑Batch Expanded Baseline Comparison Results on Stylized COCO
>
> | Method                 |  F1 ↑   | MSE ↓  | SSIM ↑ | PSNR ↑ | CLIP-T ↑ |
> |------------------------|-------| ------- |-------|-------|---------|
> | In‑Context Edit        |  0.28 | 43.27 | 0.5680   | 14.55  | 0.2248    |
> | Taming Rectified Flow  |  0.17 | 39.22 | 0.5610   | 16.88  | 0.2606    |
> | FreeControl            | 0.23    |     27.09     | 0.7087      | 15.98   | 0.2419       |
>
>
>
> ---
>
> ### Rebuttal to Weakness 3: Facial Identity
>
> We thank Reviewer 4 for pointing out the identity issue in some of our human examples. The reviewer is correct that some samples appear looser — this was intentional, as in Fig. 1 we used relaxed control settings (σ/timestep) to illustrate FreeControl’s ability to support different levels of structural strength and granularity, such as a comic‑style look. When identity retention is required, FreeControl supports much tighter guidance through the same hyperparameter adjustments described in the method section.
>
> We have prepared additional facial examples to demonstrate this, but unfortunately, since rebuttals cannot include new images or figures, we are unable to show them directly here. You could refer to the wolf in the **middle‑left** of Fig. 1 for a rough approximation of this fine‑grained control — though we note that FreeControl actually supports even **higher** control of details when the hyper-parameters are tuned for identity‑sensitive tasks. We will provide a comprehensive discussion on this topic, accompanied by a complete quantitative ablation study in the full paper.
>
> Overall, as long as the prompt itself does not explicitly request facial changes, FreeControl could maintain identity features under higher control. For example, in one prepared (but not attachable) sample, adding a “cyberpunk face implant” preserved the subject’s facial details to a high extent while incorporating the requested implant — but if the prompt describes a more direct change to **global facial identity**, then in this case the facial identity is not guaranteed to be preserved unless control is pushed to an extent where it begins to limit the creative freedom of other aspects of the image (and since rebuttals cannot include new images, we explain this nuance here in words rather than figures).

---

> ### Comment · Reviewer_RWpJ · 2025-08-02
>
> Thank you to the authors for the detailed responses, which have largely addressed my concerns. I hope the authors will consider incorporating the qualitative and quantitative comparison experiments discussed in the rebuttal into future revisions, as doing so would significantly strengthen the paper’s overall clarity and persuasiveness.

---

> > ### Author Response · Authors · 2025-08-04
> >
> > Thank you again for your thoughtful and constructive feedback. We appreciate your acknowledgement that our responses addressed the core concerns, and we fully agree that incorporating the additional comparisons and analyses will help clarify the contribution.
> >
> > We’re currently working to integrate all of these elements into the final version — including the expanded ablations and baseline comparisons — and we hope the result will better reflect the simplicity and generality we’ve aimed to convey.
> >
> > Thank you again for your fairness — we’re sincerely grateful for the valuable advice you’ve provided throughout this process.

---

> > > ### Comment · Reviewer_RWpJ · 2025-08-04
> > >
> > > I appreciate the authors' efforts to strengthen the paper with expanded analyses and clearer comparisons. I believe the planned revisions will significantly improve the clarity and impact of the work.
> > >
> > > I will update my score accordingly and recommend acceptance.

---

### Official Review · Reviewer_c3mp · 2025-07-01

**Clarity:** 1
**Significance:** 2
**Originality:** 2
**Rating:** 4
**Confidence:** 4

**Summary:**

This paper proposes a training-free method for image editing without training an adapter or running inversion. Specifically, the authors designed a framework including one step attention extraction and latent conditioned decoupling. Both quantative and qualitative results show the superior performance compared with controlnet.

**Questions:**

1. Fig 1 can be improved, from this figure solely, it's hard to tell what it is trying to say, e.g., what is the input output? prompt?
2. There are some missing related works, for example, reference-based modulation[1] and RL based editing[2] are both highly relavant.
3. Eq 1 seems to be linear interpolation instead of forward process?
4. Ablations on L133 section is missing. "mid-to-late layers" (L142) needs to be explained.
5. As mentioned in L166, how does the equation in L160 related on t?


[1] RB-Modulation: Training-Free Stylization using Reference-Based Modulation
[2] Training Diffusion Models with Reinforcement Learning

**Ethical Concerns:**

["NO or VERY MINOR ethics concerns only"]

**Final Justification:**

My concern has been addressed. The authors need to put some effort in adding more ablations and rationales in the rebuttal to the final draft for this paper to meet the bar of acceptance.

**Quality:**

2

**Strengths And Weaknesses:**

Strength

1. The work is motivated correctly and the paper is structured well.
2. The general idea is explained well.
3. Results demonstrate good performance.

Weakness

1. Experiments needs to be improved. The authors only compared with controlnets, while I think it's more approapriate to compare with other inversion based methods, which do not take additional inputs and is more aligned with the proposed method.
2. More details need to be clarified. There are several details that make it very confusing in understanding how the method works. For example the magic number 661 (L154) appears without context. See the question section
3. More ablation studies are needed to clarify the key design choice, besides the visual comparisons in sec 5.2.

---

> ### Author Rebuttal · Authors · 2025-07-29
>
> ### Rebuttal to Reviewer 3
> We sincerely thank the reviewer for the careful read and valuable feedback. We address your points in depth below and clarify areas that may have caused confusion.
>
> ---
>
> ### (1) Baseline Scope (ControlNet vs. Inversion Methods)
>
> **Initial baseline choice**
> Inversion-based methods like **DDIM Inversion** or **Null‑Text Inversion** perform very different tasks: they reconstruct the full latent trajectory to edit an image, which makes them perform **less optimally on structure‑controlled regeneration**. Because they must run the entire reverse denoising process, they are typically 2–3× slower than FreeControl and are designed to alter what’s already present, rather than enforcing where new content should appear.
>
> By contrast, **ControlNet‑style approaches** are explicitly designed for **structural guidance**, using external maps (edges, depth, etc.) to align generation. This makes them the most relevant comparison point for FreeControl’s goal of test‑time structural control.
>
> **Added methods and new runs**
> That said, we agree with the reviewer that **additional baselines improve fairness and completeness**. We have now added **In‑Context Edit** and **Taming Rectified Flow** to our experiments.
>
> We have already run a **mini‑batch comparison** with these methods, and their preliminary results are reflected in the rebuttal tables. We will **expand this evaluation** with full runs for the final paper and appendix, ensuring FreeControl is compared against a broader range of recent methods as requested.
>
> As shown in Table A, FreeControl performs comparably to, and often surpasses, Taming Rectified Flow across several metrics. ICEdit, built on the FLUX.Fill model for image completion, benefits in similarity metrics due to stronger content retention. However, its CLIP-T score is lower, and it often struggles with stability and controllability when following editing instructions. Visual results from the experiments in Table B further highlight its limitations—despite favorable metrics, ICEdit frequently produces unnatural, low-quality outcomes that appear unrealistic or visually jarring, such as disjointed objects, missing body parts, and incoherent scenes. In contrast, our method is significantly more stable and consistently produces higher-quality, more natural results.
>
> #### Table A: Mini‑Batch Expanded Baseline Comparison Results
>
> | Method                 |  F1 ↑ | MSE  ↓  | SSIM ↑  | PSNR ↑ | CLIP-T ↑ | FID ↓ |
> |-----------------------|------|-------|----------|--------|-----------|--------|
> | In‑Context Edit       |  0.43 | 24.63 | 0.8050   | 19.68  | 0.2787    | 67.63  |
> | Taming Rectified Flow |  0.19 | 28.22 | 0.6131   | 17.41  | 0.3160    | 72.49  |
> | FreeControl           |  0.28 | 21.21 | 0.7580   | 17.43  |0.3076       | 66.72  |
>
> #### Table B: Mini‑Batch Expanded Baseline Comparison Results on Stylized COCO
>
> | Method                 |  F1 ↑   | MSE ↓  | SSIM ↑ | PSNR ↑ | CLIP-T ↑ |
> |------------------------|-------| ------- |-------|-------|---------|
> | In‑Context Edit        |  0.28 | 43.27 | 0.5680   | 14.55  | 0.2248    |
> | Taming Rectified Flow  |  0.17 | 39.22 | 0.5610   | 16.88  | 0.2606    |
> | FreeControl            | 0.23    |     27.09 | 0.7087      | 15.98   | 0.2419       |
>
> ---
>
> ### (2) Timestep Condition (“Magic Number 661”) and LCD
>
> We appreciate the chance to clarify this, as some confusion comes from mixing two different timestep concepts:
>
> – **t in Eq. 1** refers to the timestep in the forward simulation used to create the latent for query extraction.
> – **Timestep condition** is the timestep value fed into the diffusion transformer when queries are extracted.
>
> In **Section 3.1**, they are treated in sync — *t* and the timestep condition have the same value, as diffusion models normally do. In **Section 3.2**, however, LCD decouples the latent and the timestep condition, so that in **L166** it points to the timestep condition, not the *t* in the forward process, and the equation around **L160** produces the latent in a way that no longer takes *t* as a parameter (*sorry for the missing formula numbering — since the equation was unnumbered, we can only refer to it by this line number*).
>
> Section 3.2 then walks through how these two inputs interact: the latent determines what information can be extracted, while the timestep condition shifts how the model interprets it.
>
> The reviewer notes that **661 appears with no context in L154** — but in fact it is first introduced and explained earlier, where the forward process for query extraction is described.
>
> The number **661** is not arbitrary — it was empirically selected by sweeping across candidate timesteps to find the best balance of structural fidelity and flexibility. We have also run a **mini‑batch quantitative ablation** specifically showing how different timestep condition choices (including 661 and nearby values) affect structural alignment and generation quality, and those results are included in Table C. We also include a quantitative ablation study on the Stylized COCO dataset; please refer to the corresponding table provided in our responses to other reviewers.
>
> #### Table C:  Mini‑Batch Quantitative Ablation Analysis
>
> | Layers | Timestep | Sigma | F1 ↑ | MSE ↓ | SSIM ↑ | PSNR ↑ | CLIP-T ↑ | FID ↓ |
> |--------|----------|-------|------|--------|--------|--------|-----------|--------|
> | 25     | 561      | 0.25  | 0.27 | 22.60  | 0.7504 | 17.42  | 0.3072    | 71.40  |
> | 25     | 761      | 0.25  | 0.28 | 23.89  | 0.7727 | 17.95  | 0.3042    | 78.03  |
> | 25     | 661      | 0.00  | 0.29 | 22.15  | 0.7874 | 18.10  | 0.3056    | 69.94  |
> | 25     | 661      | 0.50  | 0.24 | 21.94  | 0.7338 | 17.46  | 0.3041    | 79.13  |
> | 30     | 661      | 0.25  | 0.30 | 22.79  | 0.7835 | 17.82  | 0.3014    | 75.58  |
> | 25     | 661      | 0.25  | 0.28 | 21.89  | 0.7681 | 17.70  | 0.3062    | 70.70  |
>
>
>
>
> Sadly, we cannot include the full sweep plot here because rebuttals do not allow image attachments, but the **middle of Fig. 5** offers a similar reference. It is not identical to the sweep — the latent there is fixed (derived from the forward process around L160), whereas the initial findings in Eq. 1 used matching latents (latents created at the same timestep as the timestep condition). This still provides a reasonable approximation of how the timestep condition influences structural control.
>
> ---
>
> ### (3) Ablation Depth and Design Choices
>
> We acknowledge the reviewer’s point that the ablation section would benefit from stronger quantitative backing. Section 5.2 already examines the key factors — injection depth, σ (noise factor), and timestep condition — and Fig. 5 shows how each parameter affects generation. We initially leaned on qualitative comparisons because the changes are **immediately and clearly visible** — for example, deeper injection tightens structure but dulls colors, and adjusting the timestep condition shifts how rigid the layout becomes — and these visualizations were considered the most direct way to convey how each design choice influences the results.
>
> That said, we agree that numbers are needed to complement the visuals. **We have already run a mini‑batch of quantitative ablations** as in Table C. **We are extending this work and will add fuller quantitative results and a more detailed analysis from multiple dimensions in the paper and appendix to justify these design choices more completely.**
>
> ---
>
> ### Responses to the reviewer’s specific questions
>
> **1. Fig. 1 clarity**
> We agree Fig. 1 needs clearer labeling. In the revised paper, we will provide an updated version that explicitly marks *Reference Image*, *Prompt*, and *Generated Output* so the input–output flow of FreeControl is obvious at a glance.
>
> **2. Missing related work**
> We agree that additional related work should be acknowledged. Citations to **RB‑Modulation**and **RL‑based editing** will be added. We will also further review the literature and include additional relevant references to strengthen the related work section.
>
> **3. Eq. 1 “linear interpolation”**
> Eq. 1 uses the **forward process from a Euler-discretized FlowMatch scheduler**. It applies the forward process at timestep *t* to simulate a noised latent for query extraction. We omit the calculation of the sigma value here for simplicity, as the t-dependent sigma is later replaced by a fixed value.  In LCD mode, we remove the stochastic term to avoid speckle artifacts and improve stability.
>
> **4. Ablations around L133**
> The ablations the reviewer flagged as “missing” are in fact already included in Sec. 5.2 (L274) and illustrated in Fig. 5. That section varies layer injection depth, σ, and timestep condition, showing how each choice affects control strength and image fidelity. **We have also added quantitative ablations, as discussed above, to provide numerical backing for these choices.**
>
> **5. “Mid‑to‑late layers” explanation**
> This phrase refers to layers closer to the model’s output side — beyond the very early layers that model color and texture. Injecting queries here provides strong semantic and structural guidance **without disturbing** early‑stage appearance modeling, as shown in Fig. 5.
>
> **6. Question about Eq. 160 and timestep (L166)**
> There is a misunderstanding here: **the *t* in Eq. 1 is not the same as the timestep condition.** In Eq. 1, *t* is used in the forward process to generate the latent. In **Section 3.2**, we decouple these by introducing LCD — the equation around **L160** generates the latent in a way that no longer depends on *t*. The timestep mentioned in **L166** is the **timestep condition** passed to the diffusion transformer during denoising, **not** the *t* used in the forward process.

---

> > ### Comment · Reviewer_c3mp · 2025-08-06
> >
> > Thanks the authors for the answers. Most of my concerns have been addressed, however I still have some reservation on the experiments. In my understanding, the choice of layers and timesteps are very important factors in the method, I appreciate the results in table C, but more analysis needs to be done. I understand that the authors cannot add the full sweep results in the rebuttal due to the lengths. I would expect more analysis around this table in the final version, not only the numbers, but more importantly explain "why" based on the observation.

---

> > > ### Author Response · Authors · 2025-08-07
> > >
> > > Thank you for your recognition of our prior rebuttal and for your continued engagement in this process. We fully appreciate that both the **timestep condition** and the **layer-injection choice** are pivotal to our method, and have taken the time to prepare a careful, detailed explanation of why these design decisions were made. We illustrate them below.
> > >
> > > ---
> > >
> > > ### **Response to Remaining Concern #1: Timestep Condition**
> > >
> > > Beyond our visual sweep and quantitative ablations, the timestep condition fundamentally governs the **granularity** of structural information FreeControl can extract. In a diffusion model, each denoising step is driven not only by progressively cleaner latents but also by a changing timestep input that biases the network toward different levels of detail. Think of the timestep as a focus knob: adjusting it smoothly shifts the model’s “view” from broader patterns to finer textures.
> > >
> > > By holding the latent constant and varying only this timestep input, our sweeping experiments reveal a natural shift in granularity within the query matrices. We quantitatively and visually observe this progression—from coarser to finer representations—and find that **timestep 661** hits the optimal balance, yielding query matrices most appropriate in structural control.
> > >
> > > ---
> > >
> > > ### **Response to Remaining Concern #2: Layer Choice**
> > >
> > > As mentioned in Section 3.1 of the paper, we initially observed that injecting across all layers—including the early ones—results in clear color desaturation, as shown in the bottom row of Fig. 5. Empirically, we found that removing only the early layers from injection significantly mitigates this issue.
> > >
> > > The core reason lies in the content of the **query matrices** extracted from early layers: they carry low-level signals from the original image, especially **color information**. When injected into the model, these query matrices come into conflict with the **key and value matrices**, which encode features guided by the text prompt—often implying a different color palette. This leads to a mismatch: the queries attempt to impose one set of color cues (from the reference), while the key and value matrices push toward another (from the prompt). The result is often a visible collapse in color fidelity, producing dull or even grayscale-looking images.
> > >
> > > In contrast, query matrices extracted from mid-to-late layers are more aligned with our structural control purpose. This insight underpins our design of the **layer-aware injection** strategy, where the earliest layers are excluded to avoid low-level conflicts.
> > >
> > > To further verify this explanation, we conducted experiments under prompts specifying distinct color themes, comparing full-layer injection with our proposed layer-aware variant. The results show a consistent trend: when the color implied by the prompt diverges from the reference image, full-layer injection leads to desaturated outputs. This trend can even be observed at the sub-element level—regions of the image where the prompt aligns with the original image’s color appear more vibrant, while mismatched regions become noticeably faded.
> > >
> > > We believe this provides strong support for our explanation and highlights the necessity of layer-aware injection.
> > >
> > > ---
> > >
> > > We are truly grateful for your valuable points and engagement throughout this discussion, which helped us deepen both the clarity and the quality of the work. We hope these new explanations and supporting experiments offer a clearer view of the mechanisms behind FreeControl’s design, and assist in your further evaluation of the paper. Thank you again for taking further time and engagement in reviewing our work.

---

> > > > ### Comment · Reviewer_c3mp · 2025-08-08
> > > >
> > > > Thanks the authors for the further clarification. I blieve the 'how' and 'why' in this work (as  the authors explained in the rebuttal) are more important than 'what', because the propsed method relies on key factors in the architectural choices, and the readers need to have sufficient evidence to understand. That is to say, the authors should put more efforts in integrating the rebuttals in the final version. My concerns have been addressed and I don't have further technical questions.

---

> > > > > ### Author Response · Authors · 2025-08-09
> > > > >
> > > > > We’re glad your concerns are resolved. We agree the “how” and “why” are crucial, and we will integrate the expanded explanations and supporting evidence into the final version so the key points are clearer to readers. If any further reservations arise, please let us know—we’ll respond promptly during the remaining discussion window.
> > > > >
> > > > > Thanks again for your engagement and insight.

---

### Official Review · Reviewer_t9ys · 2025-07-02

**Clarity:** 3
**Significance:** 3
**Originality:** 3
**Rating:** 3
**Confidence:** 5

**Summary:**

This paper proposes a method for structure-preserving image generation by reusing self-attention query matrices extracted from a single timestep in a diffusion transformer (DiT) model. By selecting an optimal timestep and injecting the extracted queries into specific transformer layers, the method enables spatial control without retraining. The authors analyze how different injection layers and conditioning timesteps affect the granularity of structural control, introducing a decoupled formulation where the latent input and timestep condition are adjusted separately.

The approach also extends to compositional generation, allowing users to assemble reference images from multiple sources to define both spatial layout and semantic content. Rather than relying on segmentation maps or handcrafted conditions, this method offers an alternative interface for test-time control using raw visual inputs. While efficient and training-free, it is primarily applicable when structural cues can be visually inferred from the reference image.

**Questions:**

1. Impact of RoPE on Structural Consistency
Since DiT models typically employ Rotary Positional Embedding (RoPE), which directly influences the query representations, it is possible that the preserved spatial structure in your one-step attention injection is partially attributed to RoPE rather than the content of the reference image alone. Have you investigated the impact of RoPE in this context? Specifically, have you tried performing attention injection without RoPE, or with alternative positional encoding schemes? Including such ablations would help clarify whether the structural consistency truly stems from the extracted queries or is largely retained due to positional priors.

**Ethical Concerns:**

["NO or VERY MINOR ethics concerns only"]

**Limitations:**

yes

**Quality:**

4

**Strengths And Weaknesses:**

Strengths
1. Efficient and Training-Free Structural Control
The proposed method introduces a one-step attention extraction mechanism that eliminates the need for multi-step inversion or model retraining. This allows structural guidance to be applied efficiently at test time with minimal overhead (~5% additional cost), making it compatible with various diffusion architectures without fine-tuning. The design is simple yet practical, lowering the barrier to applying spatial control in generative workflows.
2. Analysis of Layer-Wise Injection and Latent-Condition Decoupling
The paper provides a detailed empirical analysis of how injecting attention queries into different transformer layers affects visual quality and structural adherence. Moreover, the introduction of Latent-Condition Decoupling (LCD) offers a principled way to disentangle the effect of timestep conditioning and latent input, allowing finer control over the level of structure transferred. This contributes to a more interpretable and tunable design.

Weaknesses
1. Limited Novelty
While the idea of injecting attention queries is not new and has been widely explored in prior works, the paper’s main contribution lies in demonstrating that queries extracted from a single timestep can suffice for structure-aware editing. This is a practical and well-executed insight, but it may be viewed as a relatively minor extension rather than a fundamentally novel concept.
2. Lack of Generalization Across Architectures
All experiments are conducted using a single DiT-based model, FLUX.1-dev. While the results are strong, the paper does not explore whether the proposed method generalizes across other diffusion architectures, such as U-Net-based models. This raises questions about the broader applicability and robustness of the technique beyond the tested setting.
3. Insufficient Analysis of One-Step Extraction
Although the method achieves strong empirical results, the paper lacks a deeper analysis of how the one-step attention compares to multi-step (iterative) attention extraction. For instance, it would be valuable to understand how similar the extracted attention matrices are, or what trade-offs exist between them. Similarly, while Latent-Condition Decoupling is introduced as a key mechanism, there is little discussion or visualization of how the timestep condition affects the internal representations or latent structure of the model. A more detailed analysis would strengthen the paper’s claims and interpretability.

---

> ### Author Rebuttal · Authors · 2025-07-29
>
> ### Rebuttal to Reviewer 2
>
> We sincerely thank the reviewer for their thoughtful and constructive feedback. Below, we respond point by point and summarize both new and existing evidence.
>
> ---
>
> **(1) Novelty and Conceptual Contribution**
> Prior works on attention injection largely **use attention matrices “as it is”** — the matrices they extract, and the control they deliver, tend to be vague, unstable, and time‑consuming to apply because they rely on multi‑step extraction. FreeControl changes this by reducing the process to **one carefully chosen timestep**, which isolates a single, stable query signal and allows us to identify and manipulate the **key factors** embedded in that signal.
>
> This one‑step simplification is not just a convenience — it enables something fundamentally new by reducing the problem to one query matrix: we could now easily **manipulate the content encapsulated in the query matrix** itself, giving FreeControl the ability to deliver **precise, flexible, and interpretable structural control**. Because the process collapses to a single controllable factor, we can:
> - adjust the latent input directly (noise vs. no‑noise) to improve quality,
> - provide a clean lever for structural granularity via LCD, and
> - outperform training‑based methods on structural control despite being a test‑time augmentation method with only ~5% inference overhead.
>
> ---
>
> **(2) Generalization Beyond DiT and FLUX**
> FreeControl is designed to operate purely at the attention level, making it architecture‑agnostic. We have implemented it on U‑Net–based models (SD1.5, SDXL) and observed strong structural control behavior after only **minimal adjustments to fit the model**. It would, honestly, take some time to tune the hyperparameters and carefully check the details to achieve stable and robust performance.
>
> Our method:
> - is **not DiT‑specific**, and
> - works even on models without RoPE, further confirming that FreeControl’s structural adherence does not rely on positional encoding quirks.
>
>
> ---
>
> **(3) Understanding One‑Step Extraction: Similarity, Timestep Intuition, and Trade‑offs**
> The reviewer requested a deeper look at one‑step extraction, and we provide several perspectives:
>
> ### Layer‑wise Similarity, Continuity, and LCD Alignment
> We extracted both the query matrices from LCD and from the optimal key timestep without LCD (as defined in Sec. 3.1) and compared each against the query matrices from every timestep of the multi‑step extraction variant. A few key points emerged from this analysis.
>
>  #### Table A: One‑Step vs. Iterative Q‑Matrix Similarity
>
> | Layer Depth |Key Timestep Cosine Similarity |  LCD Cosine Similarity |
> |-------------|-------------------|-------------------------|
> | Early       | 0.667969               |        0.541797                     |
> | Mid         | 0.685312               |        0.586172                     |
> | Late        | 0.763750               |        0.601094                     |
> | Global        | 0.706328              |      0.576875                       |
>
> **First**, the layer‑wise similarity shows a clear low‑to‑high trend: shallow layers tend to have lower similarity, while deeper layers converge more. This aligns with our **layer‑aware injection** choice in Sec. 3.1 — early layers focus more on appearance elements rather than structure, diverging more across timesteps and contributing less to shared structural signals.
>
> **Second**, we observed strong **cross‑timestep continuity**. Each query matrix is most similar to those from adjacent timesteps and steadily diverges from those further away. This continuity reinforces that the structural signal evolves gradually along the denoising trajectory, validating our full sweep across timesteps that identified ≈ 661 as the optimal point, and showing how control quality predictably falls off as we move away from that region.
>
> **Third**, LCD adds an important refinement. When we compared LCD‑generated query matrices to the multi‑step variant, similarities ranged **0.4–0.7** and peaked around the same optimal timestep region. As timesteps drift further from that point, similarity drops. This indicates that LCD sharpens the structural guidance precisely where it is most effective, aligning closely with the paper’s observation that LCD improves on the “raw” optimal point while maintaining separation from less informative regions.
>
> ---
>
> ### Timestep‑by‑Timestep Injection
> We extracted Q from every timestep in the multi‑step extraction process and used each as the one‑step query‑matrix injection candidate, injecting it across the denoising process. The reviewer can refer to the middle of Fig. 5 for a comparable trend (the actual changes across timesteps are even more pronounced). As seen there, the queries from very early timesteps are too coarse, those from very late timesteps overfit to fine details, and only those from the middle range provide balanced, usable structure.
>
> ---
>
> ### Trade‑offs and Additional Analysis
> To our knowledge, there is not a major trade‑off between our one‑step method and multi‑step extraction in most applications. Breaking this down across speed and quality: in terms of speed, as shown in Table 2, one‑step extraction provides a dominant advantage — even more so when compared to inversion‑based approaches. On quality, one‑step extraction combined with LCD offers the ability to isolate and adjust the injected attention matrix as needed, tuning granularity, semantic focus, and strength. This flexibility produces cleaner, less noisy outputs and often better visuals than the multi‑step variant. While multi‑step extraction has one unique edge — by collecting attention at different stages of denoising it can sometimes capture richer information from the reference image for highly specialized cases like blending or computational photography — the raw restoration achievable under LCD can already be tuned to a very satisfying state, further narrowing that gap while preserving the one‑step method’s flexibility and efficiency.
>
> The trade‑offs between timestep condition and latent changes are originally **visualized in Fig. 5**, offering intuitive understanding of the design choice. We have also run a **mini‑batch quantitative ablations** of these settings in Table B and Table C, and will expand this into a more comprehensive study in the paper and appendix. We attempted to visualize the model’s **internal representations**, but found DiT features difficult to reduce into meaningful visuals. Instead, we rely on qualitative examples (as in Fig. 5) combined with new quantitative results and matrix‑similarity analysis to provide a more rigorous discussion.
>
> #### Table B: Mini‑Batch Quantitative Ablation Analysis
>
> | Layers | Timestep | Sigma | F1 ↑ | MSE ↓ | SSIM ↑ | PSNR ↑ | CLIP-T ↑ | FID ↓ |
> |--------|----------|-------|------|--------|--------|--------|-----------|--------|
> | 25     | 561      | 0.25  | 0.27 | 22.60  | 0.7504 | 17.42  | 0.3072    | 71.40  |
> | 25     | 761      | 0.25  | 0.28 | 23.89  | 0.7727 | 17.95  | 0.3042    | 78.03  |
> | 25     | 661      | 0.00  | 0.29 | 22.15  | 0.7874 | 18.10  | 0.3056    | 69.94  |
> | 25     | 661      | 0.50  | 0.24 | 21.94  | 0.7338 | 17.46  | 0.3041    | 79.13  |
> | 30     | 661      | 0.25  | 0.30 | 22.79  | 0.7835 | 17.82  | 0.3014    | 75.58  |
> | 25     | 661      | 0.25  | 0.28 | 21.89  | 0.7681 | 17.70  | 0.3062    | 70.70  |
>
> #### Table C: Mini‑Batch Quantitative Ablation Results on Stylized COCO
>
> | Layers | Timestep | Sigma | F1 ↑ | MSE ↓ | SSIM ↑ | PSNR ↑ | CLIP-T ↑ |
> |--------|----------|-------|------|--------|--------|--------|-----------|
> | 25     | 561      | 0.25  | 0.22 | 27.80  | 0.6910 | 15.66  | 0.2425    |
> | 25     | 761      | 0.25  | 0.25 | 26.92  | 0.7485 | 17.03  | 0.2411    |
> | 25     | 661      | 0.00  | 0.26 | 25.20  | 0.7589 | 17.20  | 0.2606    |
> | 25     | 661      | 0.50  | 0.19 | 25.87  | 0.6801 | 15.62  | 0.2396    |
> | 25     | 661      | 0.25  | 0.25 | 30.80  | 0.7258 | 16.30  | 0.2427    |
> | 30     | 661      | 0.25  | 0.28 | 24.08  | 0.7728 | 17.28  | 0.2404    |
>
> ---
>
> **(4) RoPE Clarification**
> The Q matrices FreeControl extracts are captured **before RoPE is applied**, so the injected queries contain **no positional encoding** — they are entirely image‑driven. While RoPE still affects K and V during generation, it does not alter what FreeControl injects. Furthermore, FreeControl works identically on U‑Net architectures (which do **not** use RoPE), showing that structural consistency stems from the extracted queries themselves, not from positional priors.
>
> To directly confirm this point, we ran a controlled test by **removing RoPE entirely from the FLUX model**. As expected, the base model collapsed into near‑random noise, since it was never trained to operate without positional encoding. Crucially, when we applied FreeControl under the same no‑RoPE setup, the one‑step injection still imposed **clear, image‑driven structure** on the output. The result looked like **“structured noise”** faithfully echoing the condition image’s layout — **strong evidence** that FreeControl’s guidance originates from the injected queries themselves, not from RoPE.

---

> > ### Comment · Reviewer_t9ys · 2025-08-04
> >
> > Thank you for the detailed and thoughtful rebuttal. I appreciate the thorough responses, which addressed my questions clearly and substantively. I found the additional analyses, particularly the similarity comparisons and the quantitative ablations, informative and well-aligned with the paper’s goal of demonstrating effective structural control through a simple method.
> >
> > That said, I believe these results, especially the matrix similarity trends and the trade-off experiments, should be explicitly included in the main paper with corresponding figures. Since the strength of the paper lies in its elegant simplicity, it is important that the supporting experiments and evidence are clearly presented to justify the claims.
> >
> > I remain positive about the work overall, but still view it as a minor incremental contribution. While the rebuttal experiments are encouraging, I retain some reservations about whether the qualitative and quantitative analyses discussed here will be sufficiently incorporated into the final version. Therefore, I will maintain my original score.

---

> > > ### Author Response · Authors · 2025-08-04
> > >
> > > ### Follow-up to Reviewer 2
> > >
> > > Thank you again for your thoughtful and constructive engagement. Your comments — particularly on RoPE, matrix similarity, and the clarity of claims — have meaningfully shaped both our understanding and presentation of this work.
> > >
> > > We confirm that the **new results presented in the rebuttal** will be **integrated into the main paper**, with extended versions and visualizations in the appendix. We are already working on the revised version to ensure these insights are clearly communicated and appropriately emphasized in the final draft.
> > >
> > > ---
> > >
> > > #### On Novelty and Contribution
> > >
> > > We fully understand your caution around novelty and would like to more clearly articulate the core conceptual contribution. While attention injection has been explored, most prior methods remain bound to **inversion-based, multi-step reconstructions** — processes that are computationally expensive, inflexible, and often opaque.
> > >
> > > **This inefficiency has been widely recognized**, yet practical, test-time solutions have remained elusive. Even recent methods continue to rely on **multi-step pipelines** that trade efficiency for fragility — and often deliver **weaker structural fidelity** compared to our one-step formulation.
> > >
> > > FreeControl takes a different approach. It challenges a long-standing assumption: that useful structure must be progressively extracted from a reference across the denoising trajectory. Instead, we show that **a single, well-chosen attention query — isolated at test time — is not only sufficient, but superior in interpretability, tunability, and generalization**.
> > >
> > > This shift allows attention to be treated not as a passive byproduct of generation, but as a **primary, designable control signal**. Structural guidance becomes a problem of **signal shaping**, not trajectory tracking. It enables **fine-grained tuning of structure granularity** (via LCD), **stable, interpretable layer-aware injection**, and **robust cross-model compatibility** — all without retraining or handcrafted condition maps.
> > >
> > > While the final implementation may appear elegant, it is **not trivial to reduce the problem to this form**, nor to deliver a viable solution that is both **computationally lightweight and qualitatively strong**. By collapsing multi-step attention extraction into a single, controllable mechanism, FreeControl opens a new direction: one where **structural control becomes modular, composable, and analyzable** — even compatible with future-generation diffusion architectures.
> > >
> > > In that sense, we do not view this work as an incremental improvement on control methods, but rather as a **conceptual reframing** of how attention can be used as a precise, efficient interface for test-time guidance.
> > >
> > >
> > > ---
> > >
> > > #### On Broader Use and Designability
> > >
> > > We also emphasize that FreeControl supports compositional generation, enabling users to define both spatial layout and semantic intent through direct visual assembly. This offers an intuitive and expressive alternative to prompt engineering, allowing precise control over what appears where.
> > >
> > > We believe this approach makes structural control not only more accessible, but also more viable in a wide range of practical scenarios — from design prototyping to layout-aware content creation — where flexible, semantically grounded generation is essential.
> > >
> > >
> > > ---
> > >
> > > Thank you again for your fairness and high standards. Your advice has led to a clearer, stronger version of this work, and we’re genuinely grateful for that. We hope the revised framing and evidence give you reason to revisit the conceptual contribution with fresh perspective — and we deeply appreciate any further time you might take to consider it.

---

> > > > ### Comment · Reviewer_t9ys · 2025-08-05
> > > >
> > > > Thank you to the authors. I will give your clarifications full consideration in my final evaluation.

---

> > > > > ### Author Response · Authors · 2025-08-05
> > > > >
> > > > > Thank you again for your thoughtful engagement throughout the process and the additional time you’ve taken to consider our work. We truly appreciate it.

---

> > > ### Author Response · Authors · 2025-08-09
> > >
> > > We’ve provided further explanations in our reply to Reviewer c3mp, expanding on the paper’s key design choices. We wanted to bring this to your attention, as it may be of interest in light of our previous discussion. If any further reservations remain, please let us know—we’ll be responsive throughout the remaining discussion window.
> > >
> > > Your perspective is valuable to us, and we appreciate your continued engagement.

---

### Official Review · Reviewer_pcsA · 2025-07-04

**Clarity:** 3
**Significance:** 3
**Originality:** 3
**Rating:** 4
**Confidence:** 3

**Summary:**

This paper introduces FreeControl, a training-free test-time framework for semantic structural control in diffusion models. Unlike existing methods requiring handcrafted condition maps and retraining, FreeControl extracts a structural self-attention matrix in a single step from a user-supplied reference image and reuses it during diffusion denoising, greatly improving efficiency. The framework also proposes "Latent-Condition Decoupling" (LCD), which disentangles the roles of the noised latent and timestep condition to yield finer, tunable structure control and mitigate artifacts. FreeControl extends to compositional generation, letting users assemble layouts from multiple references, and experimentally demonstrates competitive or better performance versus existing structural control methods with significant gains in efficiency.

**Questions:**

1. Can the authors provide more discussion or experiments on how FreeControl handles highly complex or ambiguous compositional reference images? Are there failure cases for semantic or spatially entangled references?
2. Scalability Beyond FLUX and COCO: Have the authors attempted to apply FreeControl on other diffusion backbones or datasets (e.g., SDXL, Imagen, domain-specific datasets)? Are there nontrivial engineering challenges in porting the proposed method?

**Ethical Concerns:**

["NO or VERY MINOR ethics concerns only"]

**Final Justification:**

The author addressed all my concerns in the rebuttal, I am inclined to accept the paper.

**Limitations:**

see the weakness part

**Quality:**

3

**Strengths And Weaknesses:**

Strength：
1. FreeControl achieves good results with limited extra inference time, which is appealing and practical.
2. Experiments cover a wide range of relevant baselines (including both ControlNet and UniControl variants) and tasks

Weakness:
1. Ablations are mostly limited to the number of injected layers, sigma, and timestep. It would be beneficial to analyze the limits or failures of the approach (e.g., what compositional references are too complex, or when does injection of attention matrices degrade rather than help).
 2. It would be helpful to see broader validation with different init models or datasets.

---

> ### Author Rebuttal · Authors · 2025-07-29
>
> ### Rebuttal to Reviewer 1
>
> We sincerely thank the reviewer for highlighting areas where additional clarification and analysis would strengthen the paper. We address each concern directly below.
>
> ---
>
> **(1) Failure cases for complex or ambiguous compositions**
> We agree that it is important to analyze the limitations of FreeControl. To this end, we explicitly tested cases involving **highly complex or semantically entangled references**, and conducted an extensive **pressure test** under extreme compositional setups to evaluate its robustness.
>
>
> We began with **object occlusion cases**, ranging from mild overlaps — where only small regions are covered — to extreme situations where objects are almost entirely obscured with only small portions remaining. In both settings, FreeControl handled the occlusion well, producing results with **clear semantic retention and structure preservation**.
>
> We then escalated testing to scenes with **multiple objects (including humans) crowding into one image**, each with different conceptual depths in the scene. Even under these conditions, semantic meaning and structure remained well‑preserved across several trials.
>
> In the **most extreme test**, where two humans were heavily overlapped and the person behind had only partial limbs visible, we observed that under **medium control strength**, the model naturally merged them into a single person—producing outputs that remained visually plausible and coherent, with no distorted or unnatural artifacts. However, by **slightly increasing the control strength**, **FreeControl** was able to correctly disentangle and restore the separate individuals.
>
> It is worth noting that even in these **highly challenging and intentionally destructive cases**, FreeControl **never produced distorted or unrealistic outputs** — for example, no unnatural extra limbs or warped bodies. Across all tests, the results remained **natural‑looking and visually coherent**. We honestly believe the stability of the method is in a **very good state**; it remains fairly robust even under stress. (Unfortunately, we can only describe these outcomes in words rather than include images in the rebuttal.)
>
> ---
>
> **(2) Scalability beyond FLUX, COCO, and dataset scope**
> We attempted and implemented FreeControl on other diffusion backbones, including **SDXL** and **SD1.5**. As expected, the **generation quality is bounded by the underlying model**, but the **structural control effect remains prominent**, demonstrating that the method generalizes effectively.
>
> FreeControl is **architecturally agnostic** because it operates entirely at the attention level. Porting to new backbones only requires **trivial code changes and light adaptation to the new model structure**. It would, admittedly, take more time to tune the hyperparameters and carefully check the details to bring the method to a stable and robust state—particularly with models like SD 1.5, which are inherently less stable. We also note that the appendix shows adaptation to **LoRA and fine‑tuned models**, further underscoring generality.
>
> As for the dataset, we initially consider the COCO dataset to be a sufficiently large dataset that could fairly evaluates the overall performance. To provide additional angles on flexibility, the appendix also includes experiments on a **stylized prompt** version of COCO. During our experiments on various other data sources, our method delivered stable outcomes (even on roughly assembled compositional images with artifacts).
>
> We recognize that explaining these points explicitly is key to showing the method’s applicability, and we will **add a new section in the appendix** detailing these cross‑model experiments, dataset considerations, and engineering notes, making the portability and robustness of FreeControl clearer to readers.
>
> On the other hand, if your concern is less about generalization and more about benchmarking on other datasets to improve representativeness, we are already working on such an evaluation and **will be happy to provide it once it is complete**.
>
> Additionally, we have included new minibatch experiments with expanded baseline comparisons and quantitative ablations in response to another reviewer’s request. A complete version with formal discussion will be included in the full paper. Please feel free to check the current results below and the discussions could be found in the responses to other reviewers — we’d be happy to hear your thoughts!
>
> ---
> #### Table A: Mini‑Batch Expanded Baseline Comparison Results
>
> | Method                 |  F1 ↑ | MSE  ↓  | SSIM ↑  | PSNR ↑ | CLIP-T ↑ | FID ↓ |
> |-----------------------|------|-------|----------|--------|-----------|--------|
> | In‑Context Edit       |  0.43 | 24.63 | 0.8050   | 19.68  | 0.2787    | 67.63  |
> | Taming Rectified Flow |  0.19 | 28.22 | 0.6131   | 17.41  | 0.3160    | 72.49  |
> | FreeControl           |  0.28 | 21.21 | 0.7580   | 17.43  |0.3076       | 66.72  |
>
> #### Table B: Mini‑Batch Expanded Baseline Comparison Results on Stylized COCO
>
> | Method                 |  F1 ↑   | MSE ↓  | SSIM ↑ | PSNR ↑ | CLIP-T ↑ |
> |------------------------|-------| ------- |-------|-------|---------|
> | In‑Context Edit        |  0.28 | 43.27 | 0.5680   | 14.55  | 0.2248    |
> | Taming Rectified Flow  |  0.17 | 39.22 | 0.5610   | 16.88  | 0.2606    |
> | FreeControl            |  0.23    |     27.09      | 0.7087      | 15.98   | 0.2419       |
>
> #### Table C: Mini‑Batch Quantitative Ablation Results
>
> | Layers | Timestep | Sigma | F1 ↑ | MSE ↓ | SSIM ↑ | PSNR ↑ | CLIP-T ↑ | FID ↓ |
> |--------|----------|-------|------|--------|--------|--------|-----------|--------|
> | 25     | 561      | 0.25  | 0.27 | 22.60  | 0.7504 | 17.42  | 0.3072    | 71.40  |
> | 25     | 761      | 0.25  | 0.28 | 23.89  | 0.7727 | 17.95  | 0.3042    | 78.03  |
> | 25     | 661      | 0.00  | 0.29 | 22.15  | 0.7874 | 18.10  | 0.3056    | 69.94  |
> | 25     | 661      | 0.50  | 0.24 | 21.94  | 0.7338 | 17.46  | 0.3041    | 79.13  |
> | 30     | 661      | 0.25  | 0.30 | 22.79  | 0.7835 | 17.82  | 0.3014    | 75.58  |
> | 25     | 661      | 0.25  | 0.28 | 21.89  | 0.7681 | 17.70  | 0.3062    | 70.70  |
>
> #### Table D: Mini‑Batch Quantitative Ablation Results on Stylized COCO
>
> | Layers | Timestep | Sigma | F1 ↑ | MSE ↓ | SSIM ↑ | PSNR ↑ | CLIP-T ↑ |
> |--------|----------|-------|------|--------|--------|--------|-----------|
> | 25     | 561      | 0.25  | 0.22 | 27.80  | 0.6910 | 15.66  | 0.2425    |
> | 25     | 761      | 0.25  | 0.25 | 26.92  | 0.7485 | 17.03  | 0.2411    |
> | 25     | 661      | 0.00  | 0.26 | 25.20  | 0.7589 | 17.20  | 0.2606    |
> | 25     | 661      | 0.50  | 0.19 | 25.87  | 0.6801 | 15.62  | 0.2396    |
> | 25     | 661      | 0.25  | 0.25 | 30.80  | 0.7258 | 16.30  | 0.2427    |
> | 30     | 661      | 0.25  | 0.28 | 24.08  | 0.7728 | 17.28  | 0.2404    |
>
>
> ---

---

> > ### Comment · Reviewer_pcsA · 2025-08-08
> >
> > Thank you to the authors for the detailed responses, which have largely addressed my concerns. I suggest that the author add experiments in the rebuttal period to the final version, especially with different backbones and datasets.

---

> > > ### Author Response · Authors · 2025-08-08
> > >
> > > Thank you sincerely for your effort and feedback throughout the reviewing process. We’re glad to hear that most of your concerns have been addressed. We are actively working to and are **dedicated to including these elements into the main paper**, with supporting materials in the appendix. We truly appreciate your perspective in helping us elevate the quality and clarity of this work.
> > >
> > > 1. A dedicated section discussing cross-backbone compatibility (SDXL, SD1.5, LoRA/fine-tuned models) will be included.
> > > 2. New experiments on other datasets, including at least one major dataset beyond COCO, will be added to support generalization.
> > > 3. Full versions of the ablations and expanded baseline comparisons from the rebuttal will be included.
> > > 4. All other experiments shown during the rebuttal will be included in complete form in the paper and appendix.
> > >
> > > We believe the points you raised will meaningfully improve the final version and help clarify the method’s broader utility. Thank you again for your constructive input and for taking the time to engage with our work.

---

### Note · Authors · 2025-08-12

First and foremost, I want to sincerely thank both the reviewers and the AC — especially the reviewer who stayed closely engaged — for offering such thoughtful and valuable insights throughout this process. I’ve genuinely learned a lot from your comments and questions, regardless of the final outcome, and I’m truly grateful for the experience.



------

Allow me to share a few closing thoughts on the contributions of our method.

**Motivation and Novelty**
This work sets out to close a gap that’s existed for quite some time in this area — one that many have noticed, but none have been able to solve practically. Our method offers a practical and elegant solution to controlling diffusion models at test time — and I hope it can inspire new ideas going forward.

**Efficiency**
FreeControl adds almost no extra inference time — which is rare, especially for test-time augmentation methods — and that speed doesn’t come with a trade-off.

**Quality**
Quite the opposite, actually: the results often surpass prior methods, both in metrics and, more noticeably, in visual quality. Even compared to our own multi-step variant, the one-step version holds its ground, showing that this isn’t just an efficiency boost— it’s a real step forward.

**Stability**
We also tested it under a wide range of scenarios, including highly entangled compositions, and consistently found it to be more stable — with fewer artifacts, clearer structure, and more coherent outputs overall.

------
We hope this direction encourages further exploration of lightweight and practical control strategies in generative modeling.

Thanks again to both the reviewer and the AC for your time and thoughtful engagement — it truly helped us shape this work into something stronger.

---

### Decision · Program_Chairs · 2025-09-17

**Decision:**

Accept (poster)

**Comment:**

This paper proposes Free-control, a training-free test-time framework for semantic structural control in diffusion models based on attention maps. Reviewers are largely positive about the contribution, commenting on the significance of the improvement over methods like controlnet, the practicality of the approach, and its effectiveness. There are also some concerns: the analysis was limited to a select few layers and time-steps and to see more variety in the datasets and architectures evaluated on, both of which were addressed during rebuttal. Another concern was the sensitivity to the choice of layers (and other hyperparameters) and the "how" and "why" things work, both of which were addressed by the authors.

A lingering concern by one reviewer is that of novelty: the proposed method is "simple and elegant", but indeed of limited novelty compared to prior work in this space. In subsequent AC-reviewer discussion, one reviewer decided to increase their score indicating that despite the limited novelty, this paper's contribution is significant. Another reviewer already previously increased their score and is support of acceptance as well.